# Generative Social Choice: The Next Generation

**Niclas Boehmer** [1]   **Sara Fish** [2]   **Ariel D. Procaccia** [2]

## Abstract

A key task in certain democratic processes is to produce a concise slate of statements that proportionally represents the full spectrum of user opinions. This task is similar to committee elections, but unlike traditional settings, the candidate set comprises all possible statements of varying lengths, and so it can only be accessed through specific queries. Combining social choice and large language models, prior work has approached this challenge through a framework of generative social choice. We extend the framework in two fundamental ways, providing theoretical guarantees even in the face of approximately optimal queries and a budget limit on the overall length of the slate. Using GPT-4o to implement queries, we showcase our approach on datasets related to city improvement measures and drug reviews, demonstrating its effectiveness in generating representative slates from unstructured user opinions.

## 1. Introduction

In the realm of AI and democracy, one of the most widely discussed systems is *Polis* (Small et al., 2021). It enables online participants to submit statements about a given policy question and vote on the statements submitted by others. These inputs are then aggregated into a report that highlights a subset of statements seen as representative, in that they capture different salient viewpoints expressed by participants. Polis has been famously used in Taiwan, Australia, and elsewhere for national-level policymaking. A closely related system, *Remesh*, has been deployed by the United Nations for peacebuilding activities (Alavi et al., 2022).[1]

Halpern et al. (2023) observe that the task performed by these systems — selecting a representative subset of state-ments based on votes — can be viewed as a *social choice* problem. Specifically, it is an instance of *committee elections*, where the statements play the role of candidates. It is therefore possible to formalize what it means for a selection to be "representative" using powerful notions of proportionality developed in the computational social choice literature (Aziz et al., 2017a).

Our starting point is the work of Fish et al. (2024), who propose taking a broader viewpoint: instead of restricting the set of candidates to the specific statements submitted by participants, we view every possible (well-formed and reasonably concise) statement as a potential candidate. This gives rise to two related challenges when creating "representative" sets of statements: *generating* consensus statements and *predicting* the preferences of participants over newly generated statements. In practice, both of these challenges, Fish et al. (2024) argue, can be effectively addressed using large language models (LLMs).

To simultaneously tap the theoretical rigor of social choice and the (notoriously inscrutable) power of LLMs, Fish et al. (2024) put forward a two-step framework: *generative social choice*. In Step 1, they seek to design a democratic process that converts survey responses into a provably representative slate of $k$ statements, *assuming perfect answers to certain queries*, which constitute the building blocks of their process. In Step 2, they implement and validate those queries, showing that modern LLMs (specifically, GPT-4o) can reliably (albeit imperfectly) realize them. In an end-to-end pilot with hundreds of participants, the LLM-based democratic process of Fish et al. (2024) generated a slate of 5 statements that faithfully represents the views of the US population on the topic of chatbot personalization.

### 1.1. Our Contributions

In our opinion, the framework of Fish et al. (2024) provides a compelling vision for how AI can support democratic innovation. But their implementation is, at this point, largely a proof of concept; there is much more work to do before generative social choice can be put into practice. In this paper, we make progress in this direction by enhancing generative social choice along two distinct dimensions.

1. *Costs and budgets.* The number $k$ of statements put in the slate can be chosen to match the attention span of

[1]Hasso Plattner Institute, Germany [2]Harvard University, USA. Correspondence to: Niclas Boehmer <Niclas.Boehmer@hpi.de>, Sara Fish <sfish@g.harvard.edu>.

*Proceedings of the $42^{nd}$ International Conference on Machine Learning*, Vancouver, Canada. PMLR 267, 2025. Copyright 2025 by the author(s).

[1]Polis and Remesh are both examples of *collective response systems* (Ovadya, 2023).

the consumer, with larger values of $k$ allowing for the representation of more nuanced positions but imposing a greater cognitive burden. However, previously $k$ needed to be set upfront, and there was no way to control the length of statements, implying that even a small $k$ could potentially lead to a "long" slate. We view the *cost* of a statement as its length and require our slates to adhere to an overall *budget*. In this way, for example, one can request a slate with an overall length of no more than 100 words, which the algorithm can split among an arbitrary number of statements. The addition of costs and budgets to our problem is also natural in that it mirrors the extension of committee elections to *participatory budgeting* (Cabannes, 2004) — the task of selecting alternatives with costs subject to a budget constraint.

2. *Approximate queries.* As mentioned earlier, the theoretical model of Fish et al. (2024), which underlies the design of their democratic process, assumes perfect answers to certain queries. Specifically, they rely on two queries: *discriminative queries*, which predict a given participant's utility for a given statement, and *generative queries*, which (slightly reinterpreted) generate a statement maximizing the number of participants from a given subset that have at least a given utility for it. If these queries do not perform optimally, their theoretical guarantees break down.

   To address this shortcoming, we introduce approximate queries (also accounting for costs): Our discriminative query simply predicts utilities up to an additive error of $\beta$. Our generative query has three possible sources of error and hence three parameters $\gamma$, $\delta$, and $\mu$; it returns a statement that is optimal up to a multiplicative factor of $\gamma$, while relaxing the allowed cost by a factor of $\mu$ and the desired utility by an additive $\delta$ term.

Our main theoretical result is a new democratic process subject to a total budget constraint, relying on approximate queries, which guarantees approximate proportional representation. In more detail, assuming perfect answers to queries, the democratic process of Fish et al. (2024) guarantees *balanced justified representation (BJR)*, which (informally speaking) means that every subset of participants with sufficiently cohesive preferences that is large enough to deserve one statement must be represented in the output slate. Our process extends this guarantee by assuming only approximate queries and taking costs and budgets into account; we achieve an approximate version of BJR adapted to the cost setting, which gracefully degrades as a function of the error parameters $\beta, \gamma, \delta$ and $\mu$. We also establish several lower bounds, showing that the dependence of our guarantee on the error parameters is close to optimal. Notably, our process and its guarantees are agnostic to the specific

implementation of the underlying queries, and our process does not require knowledge of the magnitude of the errors in the query responses when the algorithm is executed.

In the second part of the paper, we introduce and evaluate the **Pro**portional **S**late **E**ngine (PROSE), our practical implementation of our proposed democratic process. PROSE leverages GPT-4o when answering discriminative or generative queries. In contrast to the implementation of Fish et al. (2024), PROSE is applicable to a variety of datasets, as it only requires unstructured textual user data and a target slate length as input. We evaluate PROSE on four instances drawn from drug reviews and a deliberation hosted on Polis. In each case, PROSE outperforms four baseline approaches with respect to both user satisfaction and proportionality.

Additional material, including full proofs and additional experimental results, can be found in the appendix. The code for PROSE and our other experiments is available at `github.com/sara-fish/gen-soc-choice-next-gen`.

## 1.2. Related Work

Our paper contributes to the study of leveraging AI to scale democratic processes (Kahng et al., 2019; Devine et al., 2023; Flanigan et al., 2020; Gudiño et al., 2024; Landemore, 2022). Within this literature, the emergence of LLMs has fueled efforts to create tools supporting deliberation processes at scale (Tessler et al., 2024; Bakker et al., 2022; De et al., 2025; Konya et al., 2023; Ding & Ito, 2023). Although LLMs and AI can support various stages of a deliberation process, one key challenge is identifying consensus between participants and producing a representative summary of different opinions (Small et al., 2023). A common approach in previous work has been to craft a single consensus statement that every participant endorses (Tessler et al., 2024; Bakker et al., 2022; De et al., 2025). By contrast, Fish et al. (2024) and our work aim to generate a set of (possibly conflicting) statements, with each statement representing a distinct subset of the participants. Another differentiating aspect of our contribution is the focus on deriving formal proportionality guarantees that explicitly depend on the accuracy of the underlying LLM in responding to queries.

Our work draws inspiration from and contributes to the study of committee elections in social choice (Lackner & Skowron, 2023; Faliszewski et al., 2017). Within social choice, committee elections have been extensively studied with an emphasis on defining and optimizing various measures of proportionality. This effort has resulted in a refined and well-understood hierarchy of proportionality axioms (Aziz et al., 2017a; Lackner & Skowron, 2023), to which balanced justified representation, as considered here, is in some sense orthogonal (Fish et al., 2024). Subsequent work has extended results on committee elections to participatory

budgeting, where candidates have varying costs — a model capturing, for instance, real-world applications in which a city lets voters decide on how to allocate a portion of its budget (Rey & Maly, 2023; Peters et al., 2021; Aziz & Shah, 2021). Within participatory budgeting, various preference models and ballot formats have been explored, with our model aligning closest to participatory budgeting under additive utility functions (Peters et al., 2021; Fain et al., 2016; 2018). However, unlike prior work, we do not assume a given, finite candidate set; instead, we must explore potential candidates through our query model, introducing new algorithmic and conceptual challenges.

## 2. Model

For two integers $x, y \in \mathbb{N}$, we let $[x, y] := \{x, x+1, \dots, y\}$ and $[y] := [1, y]$. Let $\mathcal{U}$ denote a (potentially infinite) universe of *statements*, and let $c : \mathcal{U} \to \mathbb{N}_0$ be a cost function mapping each statement $\alpha \in \mathcal{U}$ to its cost $c(\alpha)$. We refer to a subset of statements $W \subseteq \mathcal{U}$ as a *slate* and write its total cost as $c(W) := \sum_{\alpha \in W} c(\alpha)$. Furthermore, let $N$ be a set of $n$ agents, where each agent $i \in N$ has a utility function $u_i : \mathcal{U} \to [r]$, mapping each statement $\alpha \in \mathcal{U}$ to one of $r$ possible utility values.[2] For an agent $i \in N$, a statement $\alpha \in \mathcal{U}$, and a utility value $\ell \in [r]$, we say that $i$ *approves $\alpha$ at level $\ell$* if $u_i(\alpha) \geq \ell$. For a statement $\alpha \in \mathcal{U}$, a group of agents $S \subseteq N$, and a utility level $\ell \in [r]$, let $\sup(\alpha, S, \ell) := |\{i \in S \mid u_i(\alpha) \geq \ell\}|$ denote the number of agents from $S$ approving $\alpha$ at level $\ell$.

**General Problem Statement**  Given a budget $B \in \mathbb{N}$, our task is to select a slate $W \subseteq \mathcal{U}$ of statements of summed cost at most $B$, which proportionally represents the agents (roughly speaking, $x\%$ of the slate should represent $x\%$ of the agents; see Section 2.2). We are given the set of agents $N$ and the budget $B$, but the universe $\mathcal{U}$ and the utility functions of the agents $(u_i)_{i \in N}$ are unknown.[3] Instead, access to $\mathcal{U}$ and $(u_i)_{i \in N}$ is restricted to a set of queries (see Section 2.1). We assume that the cost $c(\alpha)$ of a statement $\alpha \in \mathcal{U}$ can be computed in constant time. In our implementation, statements will be texts, and the cost of a statement is determined by its length in words. For each agent, we are provided with textual information reflecting their opinions, and the queries will be executed with the help of LLMs.

### 2.1. Queries

We assume access to two types of queries. First, a *discriminative query* $\text{DISC}(i, \alpha)$ which takes an agent $i \in N$ and a

---

[2]Utility values can represent varying degrees of agreement with a statement, e.g., levels might correspond to "strongly approve", "approve", "indifferent", "disapprove", and "strongly disapprove".

[3]To simplify some arguments, we assume that for each $x \in [B]$, there is a statement $\alpha \in \mathcal{U}$ with $c(\alpha) = x$.

statement $\alpha \in \mathcal{U}$ as input and returns $u_i(\alpha)$. Second, a *generative query* $\text{GEN}(S, \ell, x)$, which, given a set $S \subseteq N$ of agents, a utility value $\ell \in [r]$ and a cost $x$, returns the statement approved by the largest number of agents in $S$ at level $\ell$ which costs at most $x$, i.e., $\arg\max_{\alpha \in \mathcal{U}: c(\alpha) \leq x} \sup(\alpha, S, \ell)$. We refer to these queries as *exact*.

To account for errors that naturally arise in the implementation of queries, we introduce approximate versions: For the discriminative query, we allow for an additive error. Specifically, for $\beta \in [r]$, discriminative queries are $\beta$-*accurate*, if for each agent $i \in N$ and statement $\alpha \in \mathcal{U}$, $\text{DISC}(i, \alpha) \in [u_i(\alpha) - \beta, u_i(\alpha) + \beta]$. For the generative query, we account for three types of errors: (i) a multiplicative error $\gamma$ in the number of supporters of the returned statement, (ii) a misjudgment of statements' cost by a multiplicative factor $\mu$[4], and (iii) a misjudgment of the utility agents derive from the selected statement by an additive error $\delta$. Formally, for $0 \leq \gamma, \mu \leq 1$ and $\delta \in [r]$, generative queries are $(\gamma, \delta, \mu)$-*accurate* if for the statement $\alpha^*$ returned by $\text{GEN}(S, \ell, x)$, we have $c(\alpha^*) \leq x$ and the following regarding the support of $\alpha^*$ holds:

$$\frac{\sup(\alpha^*, S, \ell - \delta)}{\max_{\alpha \in \mathcal{U}: c(\alpha) \leq \lceil \mu x \rceil} \sup(\alpha, S, \ell)} \geq \gamma \qquad (1)$$

for every $S \subseteq N$, $\ell \in [r]$, and $x \in [B]$. For $\beta = \delta = 0$ and $\mu = \gamma = 1$, we recover the exact queries.

### 2.2. Axiom

To evaluate whether a slate adequately represents the agents, we adopt an axiomatic framework following the study of proportionality in social choice. The key principle behind proportionality axioms is that a group of $x$ agents should have control over an $x/n$ fraction of the budget. A violation occurs if a group can propose a "better allocation" of their share of the budget. We introduce an approximate version of the *balanced justified representation* (BJR) axiom (Fish et al., 2024), adapted to our setting with statement costs:

**Definition 2.1** ($(b, d)$-cost**BJR**). For $b \in \mathbb{N}_0$ and $d \in \mathbb{R}_{\geq 1}$, a slate $W$ satisfies $(b, d)$-costBJR if there is a function $\omega : N \to W$ matching agents to statements in a balanced way,[5] such that no coalition $S \subseteq N$, statement $\alpha \in \mathcal{U}$ and utility threshold $\theta \in [r]$ satisfies (i) $|S| \geq d \cdot \lceil \frac{c(\alpha) \cdot n}{B} \rceil$, (ii) $u_i(\alpha) \geq \theta$ for all $i \in S$, and (iii) $u_i(\omega(i)) < \theta - b$ for all $i \in S$.

---

[4]We introduce $\mu$ because we observed that GPT-4o often undershoots the specified word budget, suggesting that it internally searches within a more conservative space (i.e., shorter statements). Consequently, we can only expect it to identify the best statement among those with length at most $z$ for some $z \leq x$. The parameter $\mu$ captures this tendency to undershoot the budget.

[5]An assignment is balanced if for each $\alpha \in W$, exactly $\lceil \frac{c(\alpha) \cdot n}{B} \rceil$ or $\lfloor \frac{c(\alpha) \cdot n}{B} \rfloor$ agents are assigned to $\alpha$. $\omega$ maps each agent to a statement *representing* the agent in the slate.

Setting $b = 0$ and $d = 1$ yields the "exact" version of the axiom, which we call *cBJR* and which can be guaranteed for correct query responses (see Theorem 3.1). Under cBJR, no coalition $S$ should exist that can control enough budget to "buy" a statement $\alpha$ for which all agents in $S$ derive utility above a threshold $\theta$, while their assigned statement in $W$ provides each of them a utility below $\theta$.[6] In the approximate version, $d$ specifies how much larger a coalition must be to constitute a violation, while $b$ quantifies how much the utility threshold $\theta$ must be preferred over the assigned statement in $W$.

If we were to drop the balancedness constraint on $\omega$, we arrive at a weaker version of the axiom to which we refer as *cost-justified representation (cJR)* in which condition (iii) becomes $u_i(\alpha') < \theta - b$ for all $\alpha' \in W$ and $i \in S$. To illustrate the differences between cJR and cBJR, consider an instance with 9 agents sharing opinion $X$ and 1 agent holding the opposite opinion $\neg X$. Suppose there are statements $\alpha_X$ (resp. $\alpha_{\neg X}$) of length $\frac{B}{10}$ that provide maximum utility for agents with opinion $X$ (resp. $\neg X$). The slate $\{\alpha_X, \alpha_{\neg X}\}$ satisfies cJR but can be seen as highly misleading, as both opinions occupy equal space in the slate despite their unequal prevalence. In contrast, cBJR ensures proportional representation: the fraction of slate space devoted to an opinion reflects its prevalence in the agent set. For instance, in this example, cBJR would require that statements covering opinion $X$ have a total length of $\frac{9B}{10}$.

At first glance, a possible limitation of cBJR is that it assumes that each agent "cares" only about one statement in the slate. However, as discussed above, larger groups of agents can still exert control over multiple statements. Moreover, as we argue in Appendix A, under certain assumptions, which are natural in our context, cBJR implies a variant of *extended justified representation (EJR)* (Peters et al., 2021), a strong proportionality axiom that explicitly models the influence of larger groups over multiple statements.

## 3. Algorithm and Theoretical Guarantees

We now introduce our new democratic process, establish its proportionality guarantees under approximate queries, and demonstrate that these guarantees are almost optimal by establishing complementary impossibility results.

Algorithm 1 presents our main parameterized algorithm. The algorithm iteratively adds statements to the slate $W$ if a sufficient number of agents "support" the statement. The outer loop (Line 2) iterates over utility levels required for

---

[6]One might wonder why we do not simply require that all agents in $S$ prefer $\alpha$ to their assigned statement in $W$, removing the utility threshold $\theta$. The resulting axiom, a version of *local stability*, has severe practical limitations: it may not always be satisfiable, and determining whether a slate satisfying the axiom exists is NP-hard (Aziz et al., 2017b).

agents to support a statement during the current round. In the inner loop (Line 4), we iterate over different possible costs of the statement to be added, where the cost values we try are given by the list $C$. For each cost value $C[j]$ and utility value $\ell$, we call the generative query, where the function $f$ controls the utility levels with which we make the call (Line 5). For each generated statement, the algorithm uses the discriminative query to identify the agents who approve the statement at level $\ell$ (Line 6). For the most approved statement $\alpha^*$ (Line 7), we check whether its approvers $S_{\alpha^*}$ can afford the statement ($|S_{\alpha^*}| \geq \lceil c(\alpha^*) \cdot n / B \rceil$; Line 8). If the condition is satisfied, $\alpha^*$ is added to the slate, and the agents that will be represented by $\alpha^*$ are removed from the population (Lines 9–10).

The runtime of the algorithm can be controlled by setting $C$ and $f$. We show in subsequent sections that: (i) Using $C = \{\lfloor j \cdot \frac{B}{n} \rfloor \mid j \in [n]\}$ and $f(\ell) = \{\ell\}$ (referred to as *Fast-DemocraticProcess*) is sufficient to achieve cBJR when queries are exact. (ii) If queries are approximate, $C = [B]$ and $f(\ell) = [\ell, r]$ (referred to as *Complex-DemocraticProcess*) leads the best guarantees. Depending on the application, the budget $B$ can be considerably larger than all other parameters, implying that the selection of $C$ might have a substantial effect on the practical running time and cost of the process.

**Differences from Fish et al. (2024)** Algorithm 1 builds on the democratic process of Fish et al. (2024) but differs in several key aspects. First, our algorithm accounts for statements with varying costs (e.g., word lengths), whereas the original process assumes uniform costs. Second, there is a fundamental difference in the generative query: While Fish et al. (2024) generate a statement that maximizes the $r$-th highest utility among a given set of agents, our query takes a utility level $\ell$ and identifies a statement approved by the largest number of agents from a given set at level $\ell$. This allows our algorithm to iteratively consider decreasing utility levels for the next statement to be added, a key feature for establishing proportionality guarantees under approximate queries, an aspect not addressed by Fish et al. (2024).

### 3.1. Warm-Up

We begin by analyzing *Fast-DemocraticProcess* (DemocraticProcess$_{C,f}$ with $C = \{\lfloor j \cdot \frac{B}{n} \rfloor \mid j \in [n]\}$ and $f(\ell) = \{\ell\}$), which satisfies cBJR when queries are exact. The key idea is that if a statement causing a cBJR violation existed, a statement with overlapping supporters would have been added to the slate. Notably, it is unnecessary to iterate over all possible cost values to rule out statements inducing cBJR violations, as the definition of cBJR depends only on the number of agents required to afford the statement (whether a statement costs $\lfloor j \cdot \frac{B}{n} \rfloor + 1$ for some $j \in [n]$ or $\lfloor (j+1) \cdot \frac{B}{n} \rfloor$ is irrelevant). In fact, the algorithm also has a

**Algorithm 1** DemocraticProcess$_{C,f}(N, B, r)$

**Parameters** List $C$ of cost values and function $f$ : $[r] \rightarrow 2^{[r]}$ mapping utility values to subsets of values.

1: $S \leftarrow N, W \leftarrow \emptyset, \ell \leftarrow r$
2: **while** $\ell \geq 1$ **and** $S \neq \emptyset$ **do**
3:     $j \leftarrow 1$
4:     **while** $B - c(W) \geq C[j]$ **and** $j \leq |C|$ **do**
5:         $U \leftarrow \bigcup_{\ell' \in f(\ell)} \{\text{GEN}(S, \ell', C[j])\}$
6:         $S_\alpha \leftarrow \{i \in S \mid \text{DISC}(i, \alpha) \geq \ell\}$ for all $\alpha \in U$
7:         $\alpha^* \leftarrow \arg\max_{\alpha \in U} |S_\alpha|$
8:         **if** $|S_{\alpha^*}| \geq \lceil c(\alpha^*) \cdot n / B \rceil$ **then**
9:             $S \leftarrow S \setminus \{\lceil c(\alpha^*) \cdot n / B \rceil$ agents $i$ from $S_{\alpha^*}$ with highest $\text{DISC}(i, \alpha^*)$ return value$\}$
10:             $W \leftarrow W \cup \{\alpha^*\}$
11:         **else**
12:             $j \leftarrow j + 1$
13:     $\ell \leftarrow \ell - 1$
14: **Return** $W$

favorable guarantee with respect to the supporter error $\gamma$ in the generative query:

**Theorem 3.1.** *For exact discriminative queries and $(\gamma, 0, 1)$-accurate generative queries, DemocraticProcess$_{C,f}$ satisfies $(0, 1/\gamma)$-cBJR if $\{\lfloor j \cdot \frac{B}{n} \rfloor \mid j \in [n]\} \subseteq C$ and $\ell \in f(\ell)$ for all $\ell \in [r]$.*

*Proof.* Let $W^*$ be the returned slate. For each $i \in N$, let $\omega(i)$ be the statement added to $W^*$ during the round in which $i$ was removed from $S$. By Lemma B.2, $\omega : N \rightarrow W^*$ and $\omega$ is balanced. For contradiction, assume there exists a group $T$ of agents witnessing a $(0, 1/\gamma)$-cBJR violation for $W^*$ at threshold $\theta$, caused by statement $\alpha'$. Let $t := \lceil \frac{c(\alpha') \cdot n}{B} \rceil$ and observe that $|T| \geq \frac{1}{\gamma} t$. Consider the first agent $\tilde{i} \in T$ removed from $S$ during the algorithm. Let $\tilde{\ell}$ and $\tilde{\alpha}^*$ be the values of the respective variables $\ell$ and $\alpha^*$ during the iteration when $\tilde{i}$ is removed from $S$. By definition, $\omega(\tilde{i}) = \tilde{\alpha}^*$ and $u_{\tilde{i}}(\tilde{\alpha}^*) \geq \tilde{\ell}$. Therefore, it suffices to show that $\tilde{\ell} \geq \theta$, contradicting $\tilde{i} \in T$.

Assume for contradiction that $\tilde{\ell} < \theta$. Consider the last iteration in which Line 4 is visited for $\ell = \theta$ and $C[j] = \lfloor t \cdot \frac{B}{n} \rfloor$ (no statement is added in this iteration). Note that as $\tilde{i}$ is the first agent from $T$ that gets removed from $S$, we have $T \subseteq S$ in this iteration, implying that the body of the loop is visited, as $|S|\frac{B}{n} \geq t\frac{B}{n} \geq C[j]$ (cf. Observation B.1). In Line 5, we will call $\text{GEN}(S, \theta, \lfloor t \cdot \frac{B}{n} \rfloor)$ and let $\zeta$ be the returned statement. As generative queries are $(\gamma, 0, 1)$-accurate, it holds that $\sup(\zeta, S, \theta) \geq t$ (since $\sup(\alpha', T, \theta) = |T| \geq \frac{1}{\gamma} t$ and $\lfloor t \cdot \frac{B}{n} \rfloor \geq \lfloor \frac{c(\alpha') \cdot n}{B} \cdot \frac{B}{n} \rfloor = c(\alpha')$). As a result, we will have $|S_{\alpha^*}| \geq t$ in Line 8 with $c(\alpha^*) \leq \lfloor t \cdot \frac{B}{n} \rfloor$, implying that

$\lceil \frac{c(\alpha^*) \cdot n}{B} \rceil \leq \lceil \frac{\lfloor t \frac{B}{n} \rfloor \cdot n}{B} \rceil \leq \lceil \frac{t \frac{B}{n} \cdot n}{B} \rceil = t \leq |S_{\alpha^*}|$. Thus, $\alpha^*$ will be added to $W$, a contradiction. $\square$

## 3.2. Approximate Queries: Guarantees

When additional sources of error ($\beta$, $\delta$, and $\mu$) are introduced into our queries, *Fast-DemocraticProcess* no longer provides (approximate) proportionality guarantees. To understand this, consider $\beta$-accurate discriminative queries for some $\beta > 0$. The issue in *Fast-DemocraticProcess* arises in Line 5, where a statement $\alpha$ is generated that is supported by agents at utility level $\ell$. However, due to errors in the discriminative query, none of these agents may end up being included in $S_\alpha$. As a result, the algorithm may ultimately produce a slate in which all agents have a utility of only $1$. In turn, with cost errors ($\mu$), we can no longer restrict our focus to querying for specific statement costs, as statements with these costs might be missed given the overestimation of costs in the generative query.

To address these challenges, we consider a more expensive version of DemocraticProcess$_{C,f}$, *Complex-DemocraticProcess*, where $C = [B]$ and $f(\ell) = [\ell, r]$. Using an extended version of the strategy from Theorem 3.1, we can establish the following bound:

**Theorem 3.2.** *For $\beta$-accurate discriminative queries and $(\gamma, \delta, \mu)$-accurate generative queries, DemocraticProcess$_{C,f}$ satisfies $(2\beta + \delta, \frac{1}{\gamma\mu})$-cBJR if $C = [B]$ and $[\ell, r] \subseteq f(\ell)$ for all $\ell \in [r]$.*

This result demonstrates the different influence that error sources have on the approximation guarantees. Upon closer examination, these dependencies become intuitive: (i) The multiplicative error $\gamma$ in the number of supporters implies that a statement $\alpha$ must be approved by $\frac{1}{\gamma} \lceil c(\alpha) \cdot n / B \rceil$ agents to ensure a statement approved by $\lceil c(\alpha) \cdot n / B \rceil$ agents is generated. (ii) For the cost error $\mu$, the generative query effectively misjudges the cost of statements by a factor of $\mu$. This requires the group of supporters to be larger by the same factor for the generative query to recognize the statement as "admissible". (iii) The discriminative error $\beta$ leads to errors in Lines 6 and 9, leading to incorrect selection of agents and a reduction in the utility agents derive from the statements they are matched to. (iv) Similarly, the utility error $\delta$ in the generative query reduces the utility agents have for the statement returned by the generative query by up to $\delta$.

## 3.3. Approximate Queries: Impossibility Results

In this section, we demonstrate that the approximate guarantees of *Complex-DemocraticProcess* cannot be significantly improved. We begin by establishing a lower bound on the impact of errors in utility judgments, which exactly matches the guarantee provided by Theorem 3.2:

**Theorem 3.3.** *Fix $\beta, \delta \in [r]$ and $\epsilon > 0$. No algorithm with*

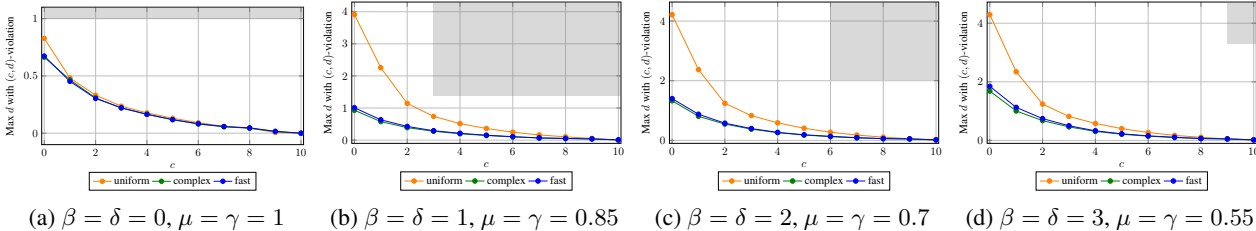

**Figure 1.** Proportionality of different versions of our democratic process for varying error settings: For a given $c$-value ($x$-axis), the plot shows the maximum $d$ value for which $(c, d)$-cBJR is violated ($y$-axis) averaged over 100 instances. The shaded gray region represents the area ruled out by the guarantees provided by Theorem 3.2 for *Complex-DemocraticProcess*.

access to $\beta$-*accurate discriminative queries and* $(1, \delta, 1)$-*accurate generative queries can guarantee* $(2\beta + \delta - \epsilon, 1)$-*cBJR [or* $(2\beta + \delta - \epsilon, 1)$-*cJR], even when all statements have unit cost.*

Turning to errors in the generative query regarding the number of supporters and cost, we establish the following impossibility result:

**Theorem 3.4.** *Fix some* $\gamma, \mu \in \mathbb{Q}_{1 > x \geq 0}$, *and* $p \in \mathbb{N}_0$. *No algorithm with access to exact discriminative queries and* $(\gamma, 0, \mu)$-*accurate generative queries can guarantee* $(p, \frac{1}{\mu} \frac{|W|}{|W|\gamma+1})$-*cBJR [or* $(p, \frac{1}{\mu} \frac{|W|}{|W|\gamma+1})$-*cJR], where $W$ is the slate returned by the algorithm.*[7]

Note that both Theorem 3.3 and Theorem 3.4 assume that certain aspects of the queries are error-free—specifically, Theorem 3.3 assumes $\gamma = \mu = 1$ and Theorem 3.4 assumes $\beta = \delta = 0$. However, allowing for additional error will never lead to an improvement in the worst case. Therefore, the above impossibility results continue to hold even when these parameters are relaxed, i.e., Theorem 3.3 holds for arbitrary $\gamma, \mu \in \mathbb{Q}_{1 > x \geq 0}$, and Theorem 3.4 holds for arbitrary $\beta, \delta \in [r]$.

Note that our algorithm from Theorem 3.2 does not precisely match the lower bound from Theorem 3.4, yet $\frac{|W|}{|W|\gamma+1} = \frac{1}{\gamma + 1/|W|}$ converges to our guarantee $\frac{1}{\gamma}$ as the number of statements in the slate $W$ increases. Intuitively, the reason why DemocraticProcess$_{C,f}$ does not match this bound lies in the condition $|S_{\alpha^*}| \geq \lceil c(\alpha^*) \cdot n / B \rceil$ in Line 8. Assume there exists a statement $\alpha$ approved by $\frac{1}{\gamma}(\lceil \frac{c(\alpha) \cdot n}{B} \rceil - 1)$ agents at some level $\ell$. Then, the generative query might only return statements approved by $\lceil \frac{c(\alpha) \cdot n}{B} \rceil - 1$ agents at level $\ell$, implying that these statements never get added, despite potentially constituting a $(0, \frac{1}{\gamma} - \epsilon)$-cBJR violation.

In the special case where all statements have unit costs, we can close this gap by adjusting the threshold in Line

8 to $\frac{n\gamma}{B\gamma+1}$. This modification addresses the above issue of large groups being potentially overlooked. It does so at the expense of allowing groups to slightly "overspend" their budget. However, this overspending is controlled so that not enough agents remain in $S$ at the algorithm's termination to constitute a cBJR violation. This adjustment allows the guarantee to match the lower bounds established in Theorems 3.3 and 3.4:

**Proposition 3.5.** *Let* $\epsilon > 0$. *Assuming all statements have unit cost, given access to* $\beta$-*accurate discriminative queries and* $(\gamma, \delta, \mu)$-*accurate generative queries, there is an algorithm (Algorithm 2 in Appendix B) that guarantees* $(2\beta + \delta, \frac{B}{B\gamma+1} - \epsilon)$-*cBJR when $\mu$ and $\gamma$ are known.*

### 3.4. Validation in Synthetic Environment

The goal of this section is to analyze how our algorithms are affected by approximate queries in simulations, thereby complementing the worst-case guarantees established in the previous parts. To be able to model errors in query answering, we consider a purely synthetic environment.

**Environment** Our environment is highly structured: There is a set $I$ of issues. For each issue, $[b]$ represents the set of possible opinions. A statement $\alpha$ contains opinions on a subset of issues, i.e., $\alpha \in [b]^{I_\alpha}$ for some $I_\alpha \subseteq I$. The cost of a statement is the number of issues it addresses, that is, $c(\alpha) = |I_\alpha|$, and the universe consists of all possible statements. Each agent $i \in N$ is characterized by a statement $\alpha^i \in [b]^I$ that covers all issues; we generate $\alpha^i$ uniformly at random. The utility of agent $i$ for statement $\alpha$ is $\sum_{j \in I_\alpha} \frac{b}{2} - |\alpha^i_j - \alpha_j|$. For efficiency reasons, we focus here on $b = |I| = 5$ and set $n = 60$ and $B = 15$.

Since the universe is finite, we can compute exact answers to both discriminative and generative queries. To simulate a $\beta$-accurate discriminative query, we draw an integer uniformly at random from $[-\beta, \beta]$ and return the true utility plus the drawn value. To simulate a $(\gamma, \delta, \mu)$-accurate generative query, we compute all statements satisfying Equation (1) and then return a statement drawn uniformly at random from this set.

---

[7]Technically, our proof shows the following: For a given instance with budget $B$, where $c$ is the minimum cost of a statement that is approved at level 2 by at least one agent and returned by the generative query, no algorithm can guarantee $(p, \frac{1}{\mu} \frac{B/c}{B/c\gamma+1})$-cBJR.

**Results**    In addition to *Fast-DemocraticProcess* (Fast) and *Complex-DemocraticProcess* (Complex), we also analyze DemocraticProcess$_{C,f}$ with $C = \{|I|\}$ and $f(\ell) = \ell$ (Uniform). This variant resembles a committee election approach and only adds statements addressing all issues. Figure 1 shows the average $(c, d)$-cBJR achieved by the slates returned by the processes. We find that all processes perform significantly better than the strongest theoretical guarantee (gray area), which holds only for Complex. Comparing algorithms, Uniform quickly incurs substantial cBJR violations even with small errors, but the violations remain relatively stable as the error increases. Fast and Complex perform similarly, with Complex slightly outperforming Fast. Both consistently outperform Uniform, but their violations increase as the error grows.

In Appendix C, we further compare the utility agents derive from their assigned statements, observing that Complex performs better than Fast, which in turn outperforms Uniform. We also provide a more detailed analysis of how increasing errors influences the proportionality achieved by the slate. Additionally, as the errors in the described environment can be interpreted as random, we examine scenarios where errors behave in a more adversarial, worst-case manner. The general trends persist, but the performance gap between Fast and Complex (as seen in Figure 1) becomes more pronounced.

# 4. Implementation and Experiments

We present the **Pro**portional **S**late **E**ngine (PROSE) and evaluate it in experiments.

## 4.1. PROSE: Proportional Slate Engine

PROSE is a practical implementation of Algorithm 1 in which both the generative and discriminative queries are implemented with the help of LLMs, in particular GPT-4o-2024-11-20. Broadly speaking, our implementation of the discriminative query uses GPT-4o as models for human preferences (similarly to the work of Fish et al. (2024), Filippas et al. (2024), and Argyle et al. (2023)). For the generative query, we note that our theoretical results are agnostic to the chosen query implementation and all theoretical results remain valid when additional statements of cost at most $C[j]$ are included in the set $U$ in Line 5 of Algorithm 1. Thus, we treat the generative query as an approximate solver for the optimization task defined in Equation (1): We use different query implementations to add multiple candidate statements to the set $U$, from which ultimately only the one with the highest number of supporters at level $\ell$ will be returned.[8]

We refer to Appendix D for a description of our query implementation and additional details on PROSE.

As input, PROSE requires the desired slate length and some textual information about each user. Importantly, PROSE can operate with unstructured and minimalistic user data, making it usable in a wide range of scenarios. This stands in contrast to the implementation of Fish et al. (2024), which relies on highly structured, curated user input. Their approach requires participants to complete detailed surveys, including rating example statements and answering predefined questions—information that is not available in our less-structured datasets. We opted not to reimplement their queries for unstructured data, as there would be substantial ambiguity in the reimplementation due to subjective implementation choices. Instead, we will include a baseline called PROSE-UnitCost, which aligns with the core unit-cost assumptions of Fish et al. (2024) but uses our own query implementations to ensure comparability.

A further advantage of PROSE is that it does not require dataset-specific tuning, e.g., we do not need to tune parameters such as the number of output statements, clustering granularity, and prompt wording. Accordingly, the four slates from our experiments are all generated using the same configuration, adding to PROSE's flexible usage.

**Limitations**    Although GPT-4o offers a powerful and flexibly usable implementation of discriminative and generative queries—leading to the generation of high-quality slates—it also comes with several limitations. Most notably, due to its opaqueness, GPT-4o comes without guarantees on the query answers, and individual responses can be very prompt-dependent, hallucinated, and potentially biased (see our impact statement for further discussion). These problems are particularly prevalent in the context of the significantly more demanding generative query, which we try to mitigate by always considering many statements for addition. During our implementation process of the generative query, we observed that GPT-4o struggles to identify coherent subgroups of agents with aligned opinions, which motivated our use of a dedicated two-step generation strategy where GPT-4o is only queried to write a consensus statement for a cohesive group of agents.

## 4.2. Experiments

In this section, we use PROSE to generate slates for four different instances and compare the results to four baselines.

---

[8]All our implementations of the generative query follow a two-step procedure. First, we identify agents $S' \subseteq S$ who are likely to approve the same statement at level $\ell$, using embeddings and clustering or nearest-neighbor techniques. Second, we prompt

GPT-4o to generate a length-bounded consensus statement for the agents in $S'$ (similarly to the work of Fish et al. 2024, Tessler et al. 2024, and Bakker et al. 2022). Moreover, rather than limiting the selection to newly generated statements in each round, we consider all previously generated statements with costs within the current target cost as candidates for addition to the slate.

| Method | Birth Control (Uniform) | | | | Birth Control (Imbalanced) | | | | Obesity | | | | Bowling Green | | | |
|---|---|---|---|---|---|---|---|---|---|---|---|---|---|---|---|---|
| | Mean | p-value | Q1 | %BJR viol. | Mean | p-value | Q1 | %BJR viol. | Mean | p-value | Q1 | %BJR viol. | Mean | p-value | Q1 | %BJR viol. |
| **PROSE** | **3.42** | - | **3.00** | **0.19** | **3.87** | - | **3.12** | **0.14** | **3.39** | - | **3.00** | **0.10** | **3.54** | - | **3.00** | **0.05** |
| PROSE-UnitCost | 3.05 | 0.011 | 2.06 | 0.31 | 3.11 | 3.1e-07 | 2.17 | 0.34 | 3.20 | 0.224 | 2.00 | 0.2 | 2.76 | 1.1e-05 | 2.02 | 0.22 |
| Clustering | 3.14 | 0.041 | 2.30 | 0.30 | 3.28 | 4.6e-05 | 2.80 | 0.23 | 3.02 | 0.016 | 2.00 | 0.17 | 2.79 | 4.5e-04 | 2.00 | 0.37 |
| Zero-Shot | 2.76 | 3.7e-06 | 2.00 | 0.65 | 3.10 | 1.0e-07 | 2.09 | 0.36 | 2.75 | 3.5e-05 | 2.00 | 0.39 | 3.06 | 0.023 | 2.00 | 0.13 |
| Contextless Zero-Shot | 2.58 | 3.1e-09 | 2.00 | 0.68 | 2.77 | 2.3e-14 | 2.01 | 0.41 | 3.15 | 0.053 | 2.73 | 0.19 | 2.54 | 1.7e-08 | 2.00 | 0.32 |

*Table 1.* Performance of PROSE and baselines. We show the mean and 25th percentile of agents' utilities. For the baselines, we show the p-value for the null hypothesis that its mean utility is the same as for PROSE. Lastly, we show the fraction of sampled statements constituting a cBJR violation.

### 4.2.1. DATASETS (see Appendix E.1 for details)

We consider two data sources. First, the publicly available UCI ML Drug Review dataset (Gräßer et al., 2018) contains web-crawled patient reviews, each accompanied by a rating on a 1–10 scale. From this dataset, we create three subsampled instances (each with 80 agents): *Birth Control (Balanced)*, which contains reviews of a birth control medication with all ratings appearing equally often; *Birth Control (Imbalanced)*, which includes only birth control reviews with extreme and central ratings, i.e., (1,2,5,9,10); and *Obesity*, which contains reviews on a obesity medication with all ratings appearing in equal frequency.[9] Each agent is represented by the text of their submitted review (excluding their rating). The budget is $B = 160$.

Second, the *Bowling Green* dataset is drawn from a public deliberation hosted on Polis (2023) regarding improvements to the city of Bowling Green, KY, US. We manually compile an instance of 41 agents to ensure a non-trivial overlap between the agents' suggestions. Each agent is represented by the full text of their submitted comments. The budget is $B = 164$, chosen to be divisible by the number of agents to avoid rounding artifacts in one of the baselines.

### 4.2.2. BASELINES (see Appendix E.2 for details)

We compare against four baselines:

*Contextless Zero-Shot.* We cast a single prompt to GPT-4o specifying the dataset's topic and the word limit. The model is asked to generate a set of opinions that proportionally reflect the general population, with the length of opinions corresponding to their prevalence.

*Zero-Shot.* Similar to *Contextless Zero-Shot*, but each agent's description is also provided to the LLM. The model is asked to produce a proportional representation of the input agents.

*Clustering.* We embed each agent using OpenAI's `text-embedding-3-large` model and then apply

---

[9]We rebalanced the ratings in the datasets because the original rating distributions are quite degenerate, e.g., in the obesity dataset, over 70% of users give a score of 9 or 10. This skewness would make the proportional summarization task quite simple, as nearly all users would support the same kind of statements.

affinity propagation clustering (Frey & Dueck, 2007), which automatically determines the number of clusters. For each computed cluster, we call an implementation of the generative query with the cluster's share of the budget as the word limit and add the returned statement representing the cluster to the slate.

*PROSE-UnitCost.* We assume that each statement has the same cost $c(\alpha) = 1$, thereby following the original idea of Fish et al. (2024). As a result, the budget describes the number of statements, $k$, to be selected. To reflect this change, in our implementation of the generative query, we no longer impose a word limit on the generated statements. Accordingly, in Algorithm 1 we set $C = [1]$, implying that when adding a statement (of arbitrary length) to the slate we now solely check whether the statement is approved by $\lfloor \frac{n}{k} \rfloor$ agents. In our experiments, we set $B = 5$. Note that the resulting slate is of unbounded length; in fact, in our experiments, the returned slates are around two to three times longer than the budget given to the other methods.

### 4.2.3. EVALUATION AND RESULTS

Ideally, evaluating the quality of a computed slate $W$ would involve asking users to rate the statements in the slate. However, this is infeasible for our datasets and beyond the scope of this paper. Consequently, we rely on a proxy. While it might seem natural to use PROSE's own discriminative query for evaluation, doing so would unfairly favor PROSE, as this would mean that PROSE selects statements to maximize the score against it will eventually be evaluated. Instead, we employ a separate chain-of-thought (CoT) implementation of the discriminative query, which assigns a score between 1 and 6 to each agent–statement pair (see Appendix D.4 for details). We refer to the scores returned as *CoT utilities* and treat them as the true, underlying utilities. To ensure an independent evaluation, the CoT implementation differs significantly from the (non-CoT) implementation used in PROSE and is considerably more expensive. Although both implementations rely on GPT-4o and aim to approximate the same underlying preferences, we show in Appendix D.5 that their outputs exhibit relatively low correlation.

For the two zero-shot baselines, we determine a maximum-weight balanced assignment $\omega : N \rightarrow W$ between agents

and statements in the slate based on the CoT utilities; for PROSE, PROSE-UnitCost, and the clustering baseline, we use the mapping produced by the methods. This setup provides the zero-shot methods with an advantage, as they leverage the CoT utilities used for evaluation. If PROSE's mapping were recomputed in the same way, the average utility would typically improve by around 15%.

We present a quantitative evaluation of the generated slates in Table 1 (all generated slates are provided in Appendix E.3). Note that the utility of user $i$ for a slate is their CoT utility for $\omega(i)$. We report the mean and 25th percentile of user utilities, along with the p-value for the null hypothesis that the average utility of a baseline method is equal to that of PROSE. Additionally, we assess the proportionality of the generated slates. To do so, we sample 100 statements from the set of statements generated by PROSE over the course of its run on a given instance and check for each statement whether it constitutes a cBJR violation under the CoT utilities.

Examining the results, we find that PROSE consistently outperforms all four baseline methods across all instances. In particular, PROSE achieves statistically significant improvements in mean agent CoT utility, ranging from $10\%$ to $40\%$ over the baselines. The improvements are even more pronounced—around $40\%$—when considering the 25th percentile of agent CoT utility, indicating that PROSE effectively represents minority opinions by allocating groups proportional control over their share of the slate. Generally speaking, as CoT utilities range from 1 to 6, the achieved utilities can be viewed as quite high, in light of the fact that each agent controls only 2 words in the drug datasets and 4 in the Bowling Green dataset. On another note, a closer examination of PROSE's outputs reveals that it successfully maps similar users to the same statement. For example, in the drug datasets, users represented by a single statement typically share similar ratings, with the exception of mid-range ratings, which occasionally appear alongside extreme ones. Regarding proportionality, slates generated by PROSE exhibit between 7 and 49 percentage points fewer cBJR violations compared to the baselines.

It is notable that despite producing substantially longer slates PROSE-UnitCost leads to inferior results. One reason for this is that in the slates generated by PROSE-UnitCost there are typically some agents that have a very low utility for their mapped statement, indicating that five statements are not sufficient to cover the opinion spectrum in its entirety. Notably, our approach circumvents the problem of picking the "right" number of statements to be added to the slate, as the algorithm will automatically split the budget between all homogeneous groups.

Lastly, we note that the runtime of PROSE is primarily driven by the response times of the LLM used in our query implementations. In our experiments, across the four datasets, PROSE used 9.6M–25.4M input and 53.5K–96.1K output tokens, with runtimes of 31–65 minutes on a single Intel i7-8565U CPU @ 1.80GHz. Given the rapidly improving inference speeds of modern LLMs, we expect these runtimes to significantly decrease in the future. By contrast, PROSE-UnitCost required around five times fewer resources: between 2.1M and 4.4M input tokens and 15.4K to 20.2K output tokens, with runtimes between 7 and 12 minutes.

## 5. Discussion

Several promising avenues for future research emerge from our work. From a theoretical perspective, it would be interesting to explore how our approach extends from the cardinal preference setting considered here to the ordinal preference setting, which is widely studied in social choice. We expect a meaningful connection between the two settings, assuming that the candidate space is sufficiently rich. From an experimental perspective, it would be most beneficial to improve the implementation of the generative query, as the performance of the current implementation starts to degrade when queried on too many agents with diverse opinions.

A long-term goal is to apply our framework to participatory budgeting elections. Since our model naturally captures participatory budgeting[10], all methodological and theoretical results carry over. However, implementing the queries in this context poses significant challenges. In particular, determining the cost of "statements", which now correspond to city improvement projects, would be nontrivial. Even more demanding, the generative query needs to propose entirely new projects that appeal to voters while staying within a predefined budget. Although such capabilities currently exceed those of GPT-4o, future advances in LLMs may make them feasible. From a methodological perspective, "traditional" participatory budgeting rules such as sequential-Phragmén (Rey & Maly, 2023), which typically iterate over all projects, could be implemented in a query-based model. However, doing so would require more complex generative queries that account for agent-specific budgets (see Appendix F for a more detailed discussion). More broadly, access to more powerful queries could also enable the satisfaction of additional, strong proportionality guarantees such as EJR+ up to any project (Brill & Peters, 2023). These considerations reveal an intricate tradeoff between the strength of guarantees and the complexity of queries used, an intriguing direction for future research.

---

[10]The universe $\mathcal{U}$ would become the set of all possible city improvement projects, and the cost function $c$ would capture the implementation cost of a project.

## Acknowledgements

Fish was supported by an NSF Graduate Research Fellowship and a Kempner Institute Graduate Fellowship. Procaccia was partially supported by the National Science Foundation under grants IIS-2147187 and IIS-2229881; by the Office of Naval Research under grants N00014-24-1-2704 and N00014-25-1-2153; and by a grant from the Cooperative AI Foundation. We thank OpenAI for providing API credits via the Researcher Access Program. We thank the anonymous ICML reviewers for their insightful reviews.

## Impact Statement

This work advances the integration of social-choice-inspired algorithms with provable fairness guarantees and the capabilities of Large Language Models (LLMs) to construct proportional and cost-constrained slates. Our approach contributes to the methodological foundations of scalable civic participation, enabling new paradigms in democratic decision-making. Although our algorithmic democratic process provides provable proportionality guarantees, the use of LLMs to rate and generate statements introduces specific risks that must be carefully addressed before deployment in real-world settings. These risks include:

- *Bias:* LLMs may favor or disfavor certain viewpoints, leading to distorted representation in the generated slates. This can occur through incorrect predictions of user utilities or the over- or under-generation of statements reflecting particular perspectives.

- *Transparency:* Although the algorithmic process itself is transparent, LLM-generated outputs are inherently opaque and may lack reliability.

- *Manipulation:* Directly inputting user comments into the LLM could expose the process to adversarial attacks, such as prompt injections.

These risks are particularly acute in political decision-making, where the consequences of biased, opaque, or manipulable outputs are particularly severe. We emphasize that our work is at most intended to inform decision making and not to automate political deliberation or decision-making. For any application, addressing these above-mentioned risks is critical to ensuring the fair, reliable, and effective application of LLM-driven democratic processes.

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

# A. cBJR and Extended Justified Representation

Peters et al. (2021) introduce the following version of extended justified representation for participatory budgeting exercises with additive utilities:

**Definition A.1** (Peters et al. (2021)). A slate $W$ satisfies extended justified representation if there is no coalition $S \subseteq N$ and set of statements $W' \subseteq \mathcal{U}$ such that (i) $\frac{|S|}{n}B \geq \sum_{\alpha \in W'} c(\alpha)$ and (ii) $\sum_{\alpha \in W'} \min_{j \in S} u_j(\alpha) > \sum_{\alpha \in W} u_i(\alpha)$ for all $i \in S$.

Condition (i) imposes that $S$ is "large" enough to afford the statements from $W'$ while the second condition imposes that each agent $i$ from $S$ is "adequately dissatisfied" with the slate $W$, i.e., the summed utility $i$ has for $W$ is smaller than the sum of the smallest utility agents from $S$ have for each project from $W'$.

**Lemma A.2.** *Assuming that $\mathcal{U}$ is closed under the union of statements, i.e., for each pair of statement $\alpha_1, \alpha_2 \in \mathcal{U}$ there exists a statement $\alpha^* \in \mathcal{U}$ with $c(\alpha^*) \leq c(\alpha_1) + c(\alpha_2)$ and $u_i(\alpha^*) \geq u_i(\alpha_1) + u_i(\alpha_2)$ for each $i \in N$, cBJR implies extended justified representation.*

*Proof.* Let $W$ be a slate that does not fulfill extended justified representations as witnessed by a coalition $S$ and a set of statements $W'$. Then, let $\alpha^* \in \mathcal{U}$ be the statement from $\mathcal{U}$ that arises from combining all statements from $W'$, i.e., $c(\alpha^*) \leq \sum_{\alpha \in W'} c(\alpha)$ and $u_i(\alpha^*) \geq \sum_{\alpha \in W'} u_i(\alpha)$ for every $i \in N$, which exists by our assumption that $\mathcal{U}$ is closed under union. Further let $\theta := \sum_{\alpha \in W'} \min_{j \in S} u_j(\alpha)$.

We claim that $S$, $\alpha^*$ and $\theta$ constitute a cBJR violation. First observe that by the definition of extended justified representation, for any $i \in S$, we have $\theta > \sum_{\alpha \in W} u_i(\alpha)$ and in particular $\theta > u_i(\omega(i))$ for any mapping $\omega : N \to W$. Further, observe that $u_i(\alpha^*) \geq \theta$ for all $i \in S$ and that $\frac{|S|}{n}B \geq \sum_{\alpha \in W'} c(\alpha) \geq c(\alpha^*)$, implying that $S$, $\alpha^*$ and $\theta$ constitute a cBJR violation. $\qquad\square$

If one follows the standard assumption that agents' utilities are additive, the assumption that $\mathcal{U}$ is closed under the union of statements feels quite natural in the natural language context studied in the paper: Two statements $\alpha_1$ and $\alpha_2$ can be simply concatenated into a statement $\alpha$. We immediately get that $c(\alpha_1) + c(\alpha_2) = c(\alpha)$ (as the cost of a statement is simply its length in words). Given that the additivity of utilities implies that $u_i(\{\alpha_1, \alpha_2\}) = u_i(\{\alpha_1\}) + u_i(\{\alpha_2\})$, it is only natural to assume that $u_i(\{\alpha\}) = u_i(\{\alpha_1\}) + u_i(\{\alpha_2\})$.

# B. Additional Proofs for Section 3

We start by showing the following invariant:

**Observation B.1.** *In Line 4 of the algorithm, it holds that $B - c(W) \geq |S|\frac{B}{n}$.*

*Proof.* For each statements $\alpha$ that gets added to $W$ over the course of the algorithm $\lceil \frac{c(\alpha) \cdot n}{B} \rceil$ agents are added to $N \setminus S$, implying $\frac{c(W) \cdot n}{B} \leq |N \setminus S|$. It follows that $c(W) \leq |N \setminus S| \cdot \frac{B}{n} = B - |S|\frac{B}{n}$, which in turn implies the invariant. $\qquad\square$

**Lemma B.2.** *If $\lfloor \frac{B}{n} \rfloor \in C$ and $1 \in f(1)$, we have $S = \emptyset$ when DemocraticProcess$_{C,f}$ terminates. For each $i \in N$, let $\omega(i)$ be the statement that has been added to the slate $W$ in the round in which $i$ was removed from $S$. We have $\omega : N \to W$ and $\omega$ is balanced.*

*Proof.* We start by arguing that $S = \emptyset$ when the algorithm terminates. For the sake of contradiction, assume that $i \in S$ when the algorithm terminates. Consider the last iteration in which Line 4 is visited for $\ell = 1$ and $C[j] = \lfloor \frac{B}{n} \rfloor$ in which by definition no statement will be added to the slate. By Observation B.1, we entered this iteration and by our technical assumption that the universe contains at least one statement $\alpha$ with $c(\alpha) = \lfloor \frac{B}{n} \rfloor$ the generative query will return some statement of cost at most $\lfloor \frac{B}{n} \rfloor$, which will be approved by $i$ at level $\ell = 1$ (as this is the lowest approval level). As $\lceil \frac{c(\alpha^*) \cdot n}{B} \rceil \leq \lceil \frac{\lfloor \frac{B}{n} \rfloor \cdot n}{B} \rceil \leq \lceil \frac{\frac{B}{n} \cdot n}{B} \rceil = 1$, the condition in Line 8 will be fulfilled and we will add a statement in this iteration, leading to a contradiction.

Let $\omega$ be the assignment induced by the creation of $W$, i.e., each agent $i$ is mapped to the statement $\alpha$ that has been added to $W^*$ in the round in which $i$ was removed from $S$. Note that $\omega$ is balanced, as only $\lceil \frac{c(\alpha) \cdot n}{B} \rceil$ agents are removed from $S$ for each statement $\alpha \in W^*$ and $S = \emptyset$ by the end of the algorithm. $\qquad\square$

**Lemma B.3.** *For $0 \leq x \leq 1$ and $y \in \mathbb{N}$, $\lceil x \lfloor \frac{y}{x} \rfloor \rceil \geq y$.*

*Proof.* For $x = 1$, the statement trivially holds. For $x < 1$, observe $\lfloor \frac{y}{x} \rfloor \geq \frac{y}{x} - 1$ and $x \lfloor \frac{y}{x} \rfloor \geq y - x$ from which we get $\lceil x \lfloor \frac{y}{x} \rfloor \rceil \geq \lceil y - x \rceil = y$ (as $x < 1$ and $y \in \mathbb{N}$). $\qquad\square$

**Theorem 3.2.** *For $\beta$-accurate discriminative queries and $(\gamma, \delta, \mu)$-accurate generative queries, DemocraticProcess$_{C,f}$ satisfies $(2\beta + \delta, \frac{1}{\gamma\mu})$-cBJR if $C = [B]$ and $[\ell, r] \subseteq f(\ell)$ for all $\ell \in [r]$.*

*Proof.* The structure of the proof is similar to the proof of Theorem 3.1. Let $W^*$ be the returned slate. For each $i \in N$, let $\omega(i)$ be the statement added to $W^*$ during the round in which $i$ was removed from $S$. By Lemma B.2, $\omega : N \to W^*$ and $\omega$ is balanced.

For contradiction, assume there exists a group $T$ of agents that witnesses a $(2\beta + \delta, \frac{1}{\gamma\mu})$-cBJR violation for $W^*$ at threshold $\theta$, caused by statement $\alpha'$, i.e., $|T| \geq \frac{1}{\gamma\mu} \cdot \lceil \frac{c(\alpha') \cdot n}{B} \rceil$, $u_i(\alpha') \geq \theta$ and $u_i(\omega(i)) < \theta - (2\beta + \delta)$ for all $i \in T$.

Consider the first agent $\tilde{i} \in T$ removed from $S$ during the algorithm. Let $\tilde{\ell}$ and $\tilde{\alpha^*}$ be the values of the respective variables $\ell$ and $\alpha^*$ during the iteration when $\tilde{i}$ is removed from $S$. We claim that it suffices to show that $\tilde{\ell} \geq \theta - (\beta + \delta)$: By definition of $\omega$ and $S_{\tilde{\alpha^*}}$, $\omega(\tilde{i}) = \tilde{\alpha^*}$ and $\text{DISC}(\tilde{i}, \tilde{\alpha^*}) \geq \tilde{\ell}$. Since discriminative queries are $\beta$-accurate, this implies that $u_{\tilde{i}}(\tilde{\alpha^*}) \geq \theta - (2\beta + \delta)$, a contradiction to $\tilde{i} \in T$.

So for the sake of contradiction, assume that $\tilde{\ell} < \theta - (\beta + \delta)$. We argue that this would imply that an agent from $T$ should have been removed from $S$ in an earlier iteration, leading to a contradiction. Consider the last iteration in which Line 4 is visited for $\ell := \theta - (\beta + \delta)$ and $C[j] := \lfloor \frac{c(\alpha')}{\mu} \rfloor$ (no statement is added to $W$ in this iteration). Note that as $\tilde{i}$ is the first agent from $T$ that gets removed from $S$, we have $T \subseteq S$ in this iteration. The body of the loop will be visited in this iteration: We have $|S| \geq |T| \geq \frac{1}{\gamma\mu} \cdot \lceil \frac{c(\alpha') \cdot n}{B} \rceil$ and in fact $|S| \geq \lceil \frac{C[j] \cdot n}{B} \rceil$ as

$$\lceil \frac{1}{\mu} \lceil \frac{c(\alpha') \cdot n}{B} \rceil \rceil \geq \lceil \frac{c(\alpha') \cdot n}{\mu B} \rceil \geq \lceil \frac{\lfloor \frac{c(\alpha')}{\mu} \rfloor \cdot n}{B} \rceil = \lceil \frac{C[j] \cdot n}{B} \rceil. \tag{2}$$

As $|S| \frac{B}{n} \geq \lceil \frac{C[j] \cdot n}{B} \rceil \frac{B}{n} \geq C[j]$, by Observation B.1, it directly follows that $B - c(W) \geq C[j]$ and thus that the while loop will be entered.

In Line 5, we will call $\text{GEN}(S, \theta, C[j])$ (as $\theta \geq \ell$). Let $\zeta$ be the returned statement. Note that $\alpha'$ is "relevant" for this generative query: $\alpha' \in \{\alpha \in \mathcal{U} \mid c(\alpha) \leq \lceil \mu \cdot C[j] \rceil\}$, as $\lceil \mu \cdot C[j] \rceil = \lceil \mu \lfloor \frac{c(\alpha')}{\mu} \rfloor \rceil \geq c(\alpha')$ by Lemma B.3. From this, as $T \subseteq S$, we get by the definition of $T$ that

$$\max_{\alpha \in \mathcal{U}: c(\alpha) \leq \lceil \mu \cdot C[j] \rceil} \sup(\alpha, S, \theta) \geq |T| \geq \frac{1}{\gamma\mu} \cdot \lceil \frac{c(\alpha') \cdot n}{B} \rceil.$$

Further, as generative queries are $(\gamma, \delta, \mu)$-accurate, it follows that $\sup(\zeta, S, \theta - \delta) \geq \gamma \frac{1}{\gamma\mu} \cdot \lceil \frac{c(\alpha') \cdot n}{B} \rceil = \frac{1}{\mu} \lceil \frac{c(\alpha') \cdot n}{B} \rceil$. Thus, there exists a set $D$ containing at least $\frac{1}{\mu} \lceil \frac{c(\alpha') \cdot n}{B} \rceil$ agents from $S$ that have utility at least $\theta - \delta$ for $\zeta$. As discriminative queries are $\beta$-accurate and $\ell = \theta - (\beta + \delta)$, it follows that $D \subseteq S_\zeta$ and thus that $|S_\zeta| \geq \frac{1}{\mu} \lceil \frac{c(\alpha') \cdot n}{B} \rceil$. As a result, we will have $|S_{\alpha^*}| \geq \frac{1}{\mu} \lceil \frac{c(\alpha') \cdot n}{B} \rceil$ and by Equation (2) $|S_{\alpha^*}| \geq \lceil \frac{C[j] \cdot n}{B} \rceil$. As $c(\alpha^*) \leq C[j]$, it follows that the if-condition in Line 8 is satisfied and $\alpha^*$ is added to $W$, a contradiction. $\qquad\square$

**Theorem 3.3.** *Fix $\beta, \delta \in [r]$ and $\epsilon > 0$. No algorithm with access to $\beta$-accurate discriminative queries and $(1, \delta, 1)$-accurate generative queries can guarantee $(2\beta + \delta - \epsilon, 1)$-cBJR [or $(2\beta + \delta - \epsilon, 1)$-cJR], even when all statements have unit cost.*

*Proof.* All statements in the universe have cost 1, implying that the budget $B$ denotes the number of statements to be picked. We set $r = 2\beta + \delta + 1$. For notational convenience, we choose the number of agents $n$ and budget $B$ so that $B$ divides $n$ (we will specify some additional constraints on the selection of $n$ and $B$ later). Let $t := \frac{n}{B}$.

The agent set will be split into two parts $X \uplus Y$ with $|X| = t$ (the splitting is to be specified later and will depend on the decisions made by the algorithm). For each $Y' \subseteq Y$, there is a statement $\alpha^*_{Y'}$ with $u_i(\alpha^*_{Y'}) = 2\beta + \delta + 1$ for all $i \in Y'$ and

$u_i(\alpha_{Y'}^*) = 1$ for all $i \notin Y'$ and $2^n$ copies of a statement $(\alpha_{Y'}^j)_{j \in [2^n]}$ with $u_i(\alpha_{Y'}^j) = 2\beta + 1$ for all $i \in Y'$ and $u_i(\alpha_{Y'}^j) = 1$ for all $i \notin Y'$. Additionally, there is a statement $\alpha_X^*$ with $u_i(\alpha_X^*) = 2\beta + \delta + 1$ for all $i \in X$ and $u_i(\alpha_X^*) = 1$ for all $i \notin X$. Lastly, there are $2^n$ copies of a statement $(\alpha_X^j)_{j \in [2^n]}$ with $u_i(\alpha_X^j) = 2\beta + 1$ for all $i \in X$ and $u_i(\alpha_X^j) = 1$ for all $i \in Y$.

We construct the generative query as follows. Let $\text{GEN}(S, \ell, x)$ for $S \subseteq N$, $\ell \in [r]$, and $x \in \mathbb{N}$ be a prompted query. If $|S \cap X| \leq \frac{|S|}{2}$, then we return a copy of the statement $\alpha_{S \setminus X}^j$. Otherwise, we return a copy of the statement $\alpha_X^j$. We return copies of statements in a way that we always return the same copy for a given input set $S$, but different copies for different input sets $S$. It is easy to see that the generative query is $(1, \delta, 1)$-accurate, as we always return a statement $\alpha$ maximizing $\sup(\alpha, S, \ell - \delta)$.

The discriminative query returns $1 + \beta$ for any agent and any of the statements returned by the generative query, resulting in a $\beta$-accurate discriminative query. In sum, from the perspective of the algorithm, all generative queries return a different statement, and all returned statements are evaluated the same by all agents.

For $(2\beta + \delta - \epsilon, 1)$-cBJR (or $(2\beta + \delta - \epsilon, 1)$-cJR to hold), $\alpha_X^*$ or a copy of $\alpha_X^j$ needs to part of the returned slate $W$. Otherwise, the set $X$ of agents constitutes a violation, as $|X| = t = \frac{n}{B}$ and $u_i(\alpha_X^*) = 2\beta + \delta + 1$ and $u_i(\alpha) = 1 < 2\beta + \delta + 1 - (2\beta + \delta - \epsilon) = 1 + \epsilon$ for all $i \in X$ and $\alpha \in W$.

Let $\mathcal{X}$ be the set of all $\binom{n}{t}$ possible size-$t$ sets of the $n$ agents. Let $\alpha_1, \ldots, \alpha_B$ be the statements that are part of $W$ and let $N_1, \ldots, N_B$ be the agent sets that were input in the generative queries to produce the respective statements. We will now argue that (under certain conditions on $n$ and $B$) we can always find a way to pick $X \in \mathcal{X}$ so that for each $i \in [B]$, $\alpha_i = \alpha_{Y'}^j$ for some $j \in [2^n]$ and $Y' \subseteq Y$. For this, for each $i \in [B]$, it needs to hold that $|N_i \cap X| \leq \frac{|N_i|}{2}$ (the *half condition*), as this implies by the construction of the generative query that $\alpha_i = \alpha_{Y'}^j$ for some $j \in [2^n]$ and $Y' \subseteq Y$. For each statement $\alpha_i$ the *half condition* rules out some sets from $\mathcal{X}$ that can no longer be picked as $X$, i.e., all $S \in \mathcal{X}$ with $|N_i \cap S| > \frac{|N_i|}{2}$. However, for certain values of $B$ and $n$, there will always be an element from $\mathcal{X}$ remaining that we can pick.

Consider as an example $B \geq 8$ and $n = 2B$. If $N_i = \{x\}$ for some $x \in [n]$, $x$ cannot be part of $X$. If $|N_i| = 2$, $X$ cannot be identical to $N_i$. If $|N_i| = 3$, $X$ cannot be one of the three size-2 subsets of $N_i$. If $|N_i| \geq 4$, the half condition becomes vacant and no sets from $\mathcal{X}$ are excluded because of this $N_i$. Thus, each $N_i$ can either rule out from $\mathcal{X}$ all sets containing some specific agent or at most three (arbitrary) sets from $\mathcal{X}$. Let $z := |\{i \in [B] \mid |N_i| = 1\}|$. Then the total number of sets from $\mathcal{X}$ that respect the half-conditions for all $N_1, \ldots, N_B$ are

$$\binom{n - z}{2} - 3(B - z) = \binom{2B - z}{2} - 3(B - z) \geq \binom{B}{2} - 3B \geq 1,$$

where the inequalities hold as $z \leq B$ and $B \geq 8$. Thus, we can always find a set $X$ that respects the half-condition for all statements selected in the slate. We set $X$ accordingly. $X$ constitutes a $(2\beta + \delta - \epsilon, 1)$-cBJR violation for the produced slate.

$\square$

**Theorem 3.4.** *Fix some $\gamma, \mu \in \mathbb{Q}_{1 > x \geq 0}$, and $p \in \mathbb{N}_0$. No algorithm with access to exact discriminative queries and $(\gamma, 0, \mu)$-accurate generative queries can guarantee $(p, \frac{1}{\mu} \frac{|W|}{|W|\gamma + 1})$-cBJR [or $(p, \frac{1}{\mu} \frac{|W|}{|W|\gamma + 1})$-cJR], where $W$ is the slate returned by the algorithm.*[11]

*Proof.* Note that $\gamma$ and $\mu$ are given and fixed. We will now construct an instance on which no algorithm can fulfill the above guarantee. Let $s, t \in \mathbb{N}$ so that $\gamma = \frac{s}{t}$ and $a, b \in \mathbb{N}$ so that $\mu = \frac{a}{b}$ (note that $t > s$ and $b > a$). In the proof, we set $r = p + 2$. However, we will only make use of two utility levels $p + 2$ and $1$. We say that an agent $i$ *approves* a statement $\alpha$ if $u_i(\alpha) = p + 2$ and *disapproves* it if $u_i(\alpha) = 1$.

In our instance, we pick $B$ so that $\frac{B}{b}$ is an integer and we set $k := \frac{B}{b}$. Further, we pick $n$ and $B$ so that $\frac{n}{B}$, $\frac{n}{k\gamma + 1}$, and $\frac{\gamma n}{k\gamma + 1}$ are integers. For notational convenience, let $g := \frac{n}{k\gamma + 1} \in \mathbb{N}$.[12] All statements in the universe have either cost $1$, $a$, or $b$. For

---

[11]Technically, our proof shows the following: For a given instance with budget $B$, where $c$ is the minimum cost of a statement that is approved at level 2 by at least one agent and returned by the generative query, no algorithm can guarantee $(p, \frac{1}{\mu} \frac{B/c}{B/c\gamma + 1})$-cBJR.

[12]Note that possible values for $n$ and $B$ fulfilling the above constraints are $B = bt$ and $n = (s + 1)tb$, as in this case $k = t$, $\frac{n}{B} = (s + 1)$, $\frac{n}{k\gamma + 1} = \frac{(s+1)tb}{s+1} = tb$ and $\frac{\gamma n}{k\gamma + 1} = tb\gamma = sb$.

---

**Algorithm 2** Uniform-Approx-DemocraticProcess($N, B, r, \mu, \gamma$)

1: $S \leftarrow N, W \leftarrow \emptyset, \ell \leftarrow r$
2: **while** $\ell \geq 1$ and $S \neq \emptyset$ and $c(W) < B$ **do**
3: $\quad U \leftarrow \bigcup_{\ell' \in [\ell, r]} \{\text{GEN}(S, \ell', \lfloor \frac{1}{\mu} \rfloor)\}$
4: $\quad S_\alpha \leftarrow \{i \in S \mid \text{DISC}(i, \alpha) \geq \ell\}$ for all $\alpha \in U$
5: $\quad \alpha^* \leftarrow \arg\max_{\alpha \in U} |S_\alpha|$
6: $\quad$ **if** $|S_{\alpha^*}| \geq \frac{n\gamma}{B\gamma+1}$ **then**
7: $\quad\quad S \leftarrow S \setminus \{\lceil \frac{n\gamma}{B\gamma+1} \rceil$ agents $i$ from $S_{\alpha^*}$ with highest $\text{DISC}(i, \alpha^*)$ return value$\}$
8: $\quad\quad W \leftarrow W \cup \{\alpha^*\}$
9: $\quad$ **else**
10: $\quad\quad \ell \leftarrow \ell - 1$
11: **Return** $W$

---

each subset $S$ of agents with $|S| \leq g$, we create a statement $\alpha_S$ and a statement $\alpha_S^*$ with costs $c(\alpha_S) = b$ and $c(\alpha_S^*) = a$ that are approved by all agents from $S$ and disapproved by everyone else. Moreover, there is a dummy statement disapproved by everyone which costs 1. Generative queries behave as follows: If a query with cost $x < b$ is cast, we return the dummy statement (note that this does not violate the $(\gamma, 0, \mu)$-accuracy of the generative query, as $\mu \cdot x < a \leq b$, implying that there is no statement $\alpha$ with an approval satisfying the constraint $\alpha \in \mathcal{U} : c(\alpha) \leq \mu \cdot x$). Otherwise $(x \geq b)$, for some group $S$ of agents given in the generative query, we pick an arbitrary subgroup $S' \subseteq S$ of size $\min(|S|, \gamma g)$ and let the generative query return the statement $\alpha_{S'}$ (which is approved by everyone from $S'$ and disapproved by everyone else and has cost $b$). Thus, all generated statements are approved by at most $\gamma g$ agents. As $g$ and $\gamma g$ are integers, the constructed queries clearly fulfill the definition of $(\gamma, 0, \mu)$-accuracy.

We will prove that no algorithm can guarantee $(p, \frac{1}{\mu}\frac{k}{k\gamma+1})$, which immediately implies the theorem, as $kb = B$. Note that on the constructed instance $(p, \frac{1}{\mu}\frac{k}{k\gamma+1})$-cBJR implies that there cannot be a group $S$ of agents of size $g$ that all disapprove the statement they are matched to in a slate returned by the algorithm: Each such group $S$ jointly approve the statement $\alpha_S^*$ of cost $a$ and $S$ is "by at least a factor of $\frac{1}{\mu}\frac{k}{k\gamma+1}$ larger" than a group that deserves a statement of cost $a$, i.e., $|S| \geq \frac{1}{\mu}\frac{k}{k\gamma+1} \cdot \lceil \frac{an}{B} \rceil = \frac{1}{\mu}\frac{k}{k\gamma+1}\frac{an}{B} = \frac{k}{k\gamma+1}\frac{bn}{B} = g$. We will show the slightly stronger statement that for each feasible slate consisting of the statements returned by the above-described generative query there are $g$ agents that disapprove all statements from the slate, which constitutes a violation of the $(p, \frac{1}{\mu}\frac{k}{k\gamma+1})$-cBJR guarantee. Notably, this would also imply that the impossibility result extends to cJR.

To see why such a group of $g$ agents always exists, note that a slate contains at most $k$ statements approved by at least one agent and that all of them will be in fact approved by at most $\gamma g$ agents. Thus, for each slate, there will be at least $n - k\gamma g$ agents that disapprove all statements from the slate (recall that $n - k\gamma g$ is an integer). It remains to argue that $n - k\gamma g \geq g$:

$$n - k\gamma g \geq g$$
$$n - k\gamma \frac{n}{k\gamma + 1} \geq \frac{n}{k\gamma + 1}$$
$$1 \geq \frac{1}{k\gamma + 1} + \frac{\gamma k}{k\gamma + 1}$$
$$1 \geq 1$$

$\qquad\qquad\qquad\qquad\qquad\qquad\qquad\qquad\qquad\qquad\qquad\qquad\qquad\qquad\qquad\qquad\qquad$ $\square$

**Proposition 3.5.** *Let $\epsilon > 0$. Assuming all statements have unit cost, given access to $\beta$-accurate discriminative queries and $(\gamma, \delta, \mu)$-accurate generative queries, there is an algorithm (Algorithm 2 in Appendix B) that guarantees $(2\beta + \delta, \frac{B}{B\gamma+1} - \epsilon)$-cBJR when $\mu$ and $\gamma$ are known.*

*Proof.* Algorithm 2 proves the proposition. Let $W^*$ be the returned slate. For each $i \in N$, let $\omega(i)$ be the statement added to $W^*$ during the round in which $i$ was removed from $S$. The resulting assignment is balanced as $\frac{n\gamma}{B\gamma+1} \leq \frac{n}{B}$. Agents that remain in $S$ at the end of the algorithm are mapped to arbitrary statements while maintaining the balancedness of $\omega$.

| Errors | | | | Average Utility | | | Average 10th-Percentile Utility | | | #instances w. cBJR violation | | |
|---|---|---|---|---|---|---|---|---|---|---|---|---|
| $\beta$ | $\mu$ | $\gamma$ | $\delta$ | Uniform | Fast | Complex | Uniform | Fast | Complex | Uniform | Fast | Complex |
| 0.0 | 1.0 | 1.0 | 0.0 | 4.56 | 4.49 | 4.49 | 1.33 | 1.43 | 1.51 | 31.0 | 0.0 | 0.0 |
| 1.0 | 0.85 | 0.85 | 1.0 | 3.36 | 3.86 | 4.26 | 0.05 | 0.8 | 0.98 | 98.0 | 65.0 | 45.0 |
| 2.0 | 0.7 | 0.7 | 2.0 | 2.96 | 3.15 | 3.44 | 0.01 | 0.33 | 0.53 | 99.0 | 98.0 | 100.0 |
| 3.0 | 0.55 | 0.55 | 3.0 | 2.79 | 2.62 | 2.95 | -0.04 | 0.13 | 0.22 | 99.0 | 100.0 | 100.0 |

*Table 2.* Metrics on produced slate $W$ and mapping $\omega$ for three versions of our democratic process under varying errors.

For the sake of contradiction assume that there is a group $T$ of agents witnessing a violation of $(2\beta + \delta, \frac{B}{B\gamma+1} - \epsilon)$-cBJR for $W^*$ at threshold $\theta$, caused by statement $\alpha'$, i.e., $|T| \geq (\frac{B}{B\gamma+1} - \epsilon) \cdot \lceil \frac{n}{B} \rceil$, $u_i(\alpha') \geq \theta$ and $u_i(\omega(i)) < \theta - (2\beta + \delta)$ for all $i \in T$. We make a case distinction:

**Some agent from $T$ gets removed from $S$ throughout the algorithm**  Consider the first agent $\tilde{i} \in T$ removed from $S$ during the algorithm. Let $\tilde{\ell}$ and $\tilde{\alpha^*}$ be the values of the respective variables $\ell$ and $\alpha^*$ during the iteration when $\tilde{i}$ is removed from $S$.

We claim that it suffices to show that $\tilde{\ell} \geq \theta - (\beta + \delta)$: By definition of $\omega$ and $S_{\tilde{\alpha^*}}$, $\omega(\tilde{i}) = \tilde{\alpha^*}$ and $\mathrm{DISC}(\tilde{i}, \tilde{\alpha^*}) \geq \tilde{\ell}$. Since discriminative queries are $\beta$-accurate, this implies that $u_{\tilde{i}}(\tilde{\alpha^*}) \geq \theta - (2\beta + \delta)$, a contradiction to $\tilde{i} \in T$.

So, for the sake of contradiction, assume that $\tilde{\ell} < \theta - (\beta + \delta)$. We argue that this would imply that an agent from $T$ should have been removed from $S$ in an earlier iteration, leading to a contradiction. Consider the last time in which Line 2 is visited for $\ell := \theta - (\beta + \delta)$ (no statement is added to $W$ in this iteration). Note that as $\tilde{i}$ is the first agent from $T$ that gets removed from $S$, we have $T \subseteq S$ in this iteration. The inner part of the while-loop will be entered in this iteration, as $\tilde{i}$ will get removed from $S$ at level $\tilde{\ell} < \ell$, implying that we have $\ell > 1$ and $c(W) < B$ in this iteration.

As part of this iteration of the while-loop in Line 3, we will call $\mathrm{GENC}(S, \theta, \lfloor \frac{1}{\mu} \rfloor)$ (as $\theta \geq \ell$). Let $\zeta$ be the statement returned by this query. Note that $\alpha'$ is "relevant" for this generative query: $\alpha' \in \{\alpha \in \mathcal{U} \mid c(\alpha) \leq \lceil \mu \cdot \lfloor \frac{1}{\mu} \rfloor \rceil\}$ by Lemma B.3. As $T \subseteq S$, we know by the definition of $T$ that

$$\max_{\alpha \in \mathcal{U}: c(\alpha) \leq \lceil \mu \cdot \lfloor \frac{1}{\mu} \rfloor \rceil} \sup(\alpha, S, \theta) \geq |T| \geq \frac{B}{B\gamma+1} \cdot \lceil \frac{n}{B} \rceil.$$

As generative queries are $(\gamma, \delta, \mu)$-accurate, it follows that $\sup(\zeta, S, \theta - \delta) \geq \frac{\gamma B}{B\gamma+1} \cdot \lceil \frac{n}{B} \rceil$. Thus, there exists a set $D$ of at least $\frac{\gamma B}{B\gamma+1} \cdot \lceil \frac{n}{B} \rceil \geq \frac{n\gamma}{B\gamma+1}$ agents from $S$ that have utility at least $\theta - \delta$ for $\zeta$. As discriminative queries are $\beta$-accurate and $\ell = \theta - (\beta + \delta)$, it follows that $D \subseteq S_\zeta$ and thus that $|S_\zeta| \geq \frac{n\gamma}{B\gamma+1}$. As a result, we will have $|S_{\alpha^*}| \geq \frac{n\gamma}{B\gamma+1}$ implying that the if-condition in Line 8 is satisfied and $\alpha^*$ is added to $W$, a contradiction.

**No agent from $T$ gets removed from $S$ throughout the algorithm**  It remains to argue why it cannot be the case that no agent from $T$ gets deleted from $S$ over the course of the algorithm. Note that the above argument implies that some agent from $T$ gets deleted from $S$, if we reach level $\ell = \theta - (\beta + \gamma) - 1$. For this not to happen, we need to add $B$ statements to $W$ before reducing $\ell$ to $\theta - (\beta + \gamma) - 1$. We claim that when adding $B$ statements to the slate $W$, the algorithm needs to delete at least one agent from $T$. For this, observe that for each statement we delete $\lceil \frac{n\gamma}{B\gamma+1} \rceil$ agents. Thus, $B\lceil \frac{n\gamma}{B\gamma+1} \rceil$ agents get deleted. It remains to prove that $n - B\lceil \frac{n\gamma}{B\gamma+1} \rceil \leq \frac{B}{B\gamma+1} \cdot \lceil \frac{n}{B} \rceil$, implying that less than $|T| = (\frac{B}{B\gamma+1} - \epsilon) \cdot \lceil \frac{n}{B} \rceil$ agents remain at the end of the algorithm:

$$\frac{B}{B\gamma+1} \cdot \lceil \frac{n}{B} \rceil + B\lceil \frac{n\gamma}{B\gamma+1} \rceil \geq \frac{B}{B\gamma+1} \cdot \frac{n}{B} + B\frac{n\gamma}{B\gamma+1} = \frac{n}{B\gamma+1} + \frac{nB\gamma}{B\gamma+1} = n(\frac{B\gamma+1}{B\gamma+1}) = n.$$

$\square$

## C. Additional Material for Section 3.4

**Utility**  We show in Table 2 some metrics regarding the utility agents have for their matched statement in the slate in our experiments in the synthetic environment. We observe that while all three variants perform similarly under no errors or very

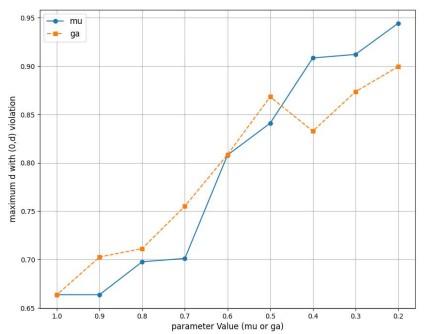
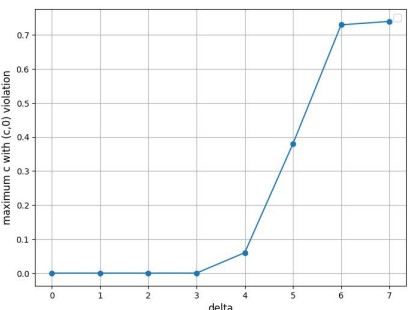
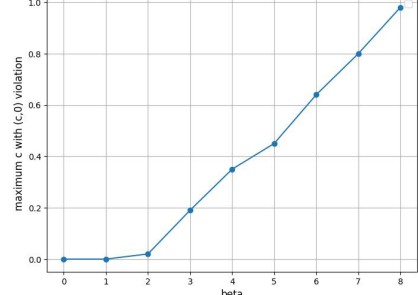

(a) Average maximum $d$ with $(0, d)$-cBJR violation when varying parameters $\mu$ and $\gamma$.

(b) Average maximum $c$ with $(c, 1)$-cBJR violation when varying parameter $\delta$.

(c) Average maximum $c$ with $(c, 1)$-cBJR violation when varying parameter $\beta$.

*Figure 2.* Comparison of proportionality violations introduced when only modifying one error parameter and keeping the others accurate.

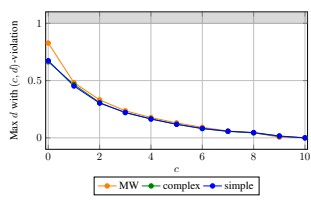
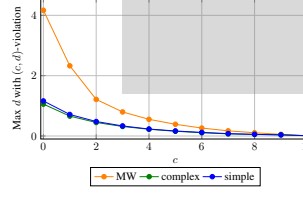
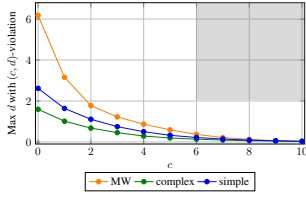
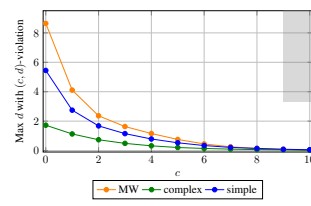

(a) $\beta = \delta = 0, \mu = \gamma = 1$  (b) $\beta = \delta = 1, \mu = \gamma = 0.85$  (c) $\beta = \delta = 2, \mu = \gamma = 0.7$  (d) $\beta = \delta = 3, \mu = \gamma = 0.55$

*Figure 3.* Analogous to Figure 1 but with worst-case error model.

| Errors | | | | Average Utility | | | Average 10th-Percentile Utility | | | #instances w. cBJR violation | | |
|---|---|---|---|---|---|---|---|---|---|---|---|---|
| $\beta$ | $\mu$ | $\gamma$ | $\delta$ | MW | Simple | Complex | MW | Simple | Complex | MW | Simple | Complex |
| 0.0 | 1.0 | 1.0 | 0.0 | 4.56 | 4.49 | 4.49 | 1.33 | 1.43 | 1.51 | 31.0 | 0.0 | 0.0 |
| 1.0 | 0.85 | 0.85 | 1.0 | 3.17 | 3.73 | 4.01 | 0.0 | 0.8 | 0.9 | 100.0 | 81.0 | 68.0 |
| 2.0 | 0.7 | 0.7 | 2.0 | 1.81 | 1.97 | 2.73 | -0.28 | -0.21 | 0.36 | 100.0 | 100.0 | 100.0 |
| 3.0 | 0.55 | 0.55 | 3.0 | 0.79 | 0.88 | 2.46 | -0.61 | -1.13 | 0.18 | 100.0 | 100.0 | 100.0 |

*Table 3.* Analogous to Table 2 but with worst-case error model.

high errors, differences emerge in the medium-error regime. Specifically, `Complex` starts to outperform `Fast`, which in turn performs better than `Uniform`.[13] A similar trend appears when taking a more egalitarian perspective and examining the average utility of agents in the bottom 10th percentile. Here, the relative differences between the three variants are more pronounced. With respect to cBJR, `Uniform` already violates cBJR in one-third of the instances even when queries are exact. When query errors are introduced, cBJR violations also quickly emerge for slates produced by the other two variants, with `Complex` showing slightly better performance for small errors.

**Scaling in Different Parameters** In Figure 2, we analyze how `Complex` behaves when increasing one error source while keeping the others at the accurate level.

**Worst-Case Error Model** In addition to the error model described in the main body, we also explored a more worst-case-focused approach. For discriminative queries, we always add or subtract $\beta$ from the true utility at random. For the generative query, we sample from the statements that achieve the worst possible ratio in Equation (1). Results can be found in Figure 3 and Table 3.

---

[13]Note that our processes are not designed to maximize average utility. For example, a slate consisting of a single long statement covering the majority opinion may achieve high average utility while severely violating proportionality.

# D. PROSE: Implementation Details

## D.1. Implementation of Discriminative Query

One key challenge in implementing discriminative queries in our setting is calibrating the evaluation of statements with varying lengths and levels of detail. Specifically, we observed that LLMs tend to rate short, generic statements more favorably. To address this, we compute two separate scores (both on a scale from 1 to 6): `Agreement`, which measures the extent to which the user agrees with the details of the statement, and `specificity`, which measures the extent to which all details from the user's description are reflected in the statement. The final utility score is computed as: $\texttt{agreement} - \texttt{specificity coefficient} \cdot (6 - \texttt{specificity})/5$, where `specificity coefficient` is a hyperparameter that allows users to influence the desired level of detail in the produced slate.[14] To calculate the `agreement` and `specificity` scores, we pass the statement and user description, along with a carefully designed description of the scoring criteria, as part of a prompt to the LLM. The final scores are derived as weighted averages of the LLM's log probabilities over its output tokens for this prompt.

**System Prompt for Agreement**

```
Your task is to determine how much a user would agree with some statement. You
will be given information about the user's opinions. Respond with a number
between 1 and 6.
```

**Prompt for Agreement**

```
Rating scale meaning:
{6: "Perfect match. User agrees 100% with all details in the statement.",
5: "Near-perfect match. User agrees with 95-99% of the statement, with only minor
discrepancies.",
4: "Substantial agreement. User agrees with about 70-80% of the statement's
content.",
3: "Moderate agreement. User agrees with roughly half of the statement\'s
points.",
2: "Minimal agreement. User agrees with only a small portion (about 20-30%) of
the statement.",
1: "No agreement. Statement contradicts or is irrelevant to user's opinion."}

Statement:
<insert statement>

User information:
<insert user information>

Now it is time for you to give the most accurate numerical rating. Write a number
between 1 and 6 and nothing else. Your estimated rating:
```

**System Prompt for Specificity**

```
Your task is to determine how much detail a statement has. You will be given a
statement, and information about the user's opinions. Your task is to determine
how many *specific details* from the user's opinion are present in the statement.
Respond with a number between 1 and 6.
```

**Prompt for Specificity**

---

[14]In our experiments, we set `specificity coefficient` = 1.

```
Rating scale meaning:
{6: "Exhaustive. All specific details from user's opinion are present in the
statement.",
5: "Nearly complete. Statement contains most specific details from user's
opinion, with only minor omissions.",
4: "Partially detailed. About 70-80% of specific details from user's opinion are
included.",
3: "Moderately specific. Roughly half of the specific details in the user's
opinion are mentioned.",
2: "Minimally specific. Only a few (20-30%) of the specific details in the user's
opinion are included.",
1: "Non-specific. Statement is entirely general, without any specific details
from user's opinion.",}

Statement:
<insert statement>

User information:
<insert user information>

Now it is time for you to give the most accurate numerical rating. Write a number
between 1 and 6 and nothing else. Your estimated rating:
```

### D.2. Implementation of Generative Query

We employ multiple strategies to generate statements. We refer to each strategy as a generator. All generators consist of two stages. In the first stage, we identify groups of agents with similar opinions. In the second stage, for a computed group of similar agents, we pass the descriptions of all users in the group to the LLM and prompt it to generate a statement of at most a given length that accurately reflects the opinion of all users in the group.

#### D.2.1. EMBEDDINGS

We consider two types of embeddings. First, we embed each agent using their description via OpenAI's `embedding-3-large`.

The computation of the second embedding is slightly more involved. We first use LLM queries to create a list of brief statements covering all issues or opinions raised by any user. Then we subsample $50$ of these statements and apply a very cheap variant of the discriminative query to compute agents' agreement with the statements on the following scale: · 1: Strongly goes against user's opinion · 2: Goes against user's opinion · 3: Somewhat goes against user's opinion · 4: Neutral / unknown · 5: Somewhat aligned with user's opinion · 6: Aligned with user's opinion · 7: Strongly aligned with user's opinion The computed ratings will serve as our embedding.

#### D.2.2. FINDING "COHESIVE" GROUPS OF AGENTS

Given an embedding of the agents, we employ four different strategies to identify cohesive groups of agents given a call of the generative query $\text{GEN}(S, \ell, x)$:

**TagNNGenerator** We select an agent at random from $S$ and compute the $\lceil \frac{x \cdot n}{B} \rceil$ agents from $S$ with the smallest Euclidean distance to the agent in the embedding.

**WeightedNNGenerator** For each agent from $S$, we compute the $\lceil \frac{x \cdot n}{B} \rceil$ agents from $S$ with the smallest Euclidean distance to the agent in the embedding. Then, we assign probabilities to these generated clusters so that each agent has a similar probability of being included in the sampled cluster. We draw a sample with these probabilities and return the sampled cluster.

**ClosestClusterGenerator** For each agent from $S$, we create a cluster of $\lceil \frac{x \cdot n}{B} \rceil$ agents by starting with the cluster only containing the agent and iteratively adding agents from $S$ to minimize the resulting summed Euclidean distance

between agents in the cluster. We sample a generated cluster with probability anti-proportional to the summed distance between agents in the cluster.

**PreviousBestGenerator** We go through all previously generated statements and compute for each of them the agents from $S$ approving the statement at level $\ell$. We return the biggest so-computed group.

### D.2.3. GENERATING STATEMENTS

Once we have identified a group of agents, we prompt the LLM with their descriptions as part of the following prompt to generate a consensus statement for this group:

**System Prompt for Consensus Statement**

```
You will be given information about a group of users and their thoughts on a
topic. Your task is to write a short, strong opinion that reflects a single,
clear stance reflecting the users' thoughts. The opinion should sound personal,
direct, and conversational, as if written by someone expressing their own
thoughts. Avoid summary-style language or listing multiple viewpoints. The level
of detail included in the opinion you write should be VERY HIGH. Use at most
<word budget> words. Write the opinion in XML tags <opinion>...</opinion>.
```

**Prompt for Consensus Statement**

```
User information:
<list of user information for all agents given as input to generative query>

Now write the opinion.
```

### D.3. Other Implementation Details of PROSE

**Generators**   For each cost value and approval level we iterate over, we use each of the four generators twice, where for TagNN, WeightedNN, and ClosestCluster we use each embedding once.

**Cost Values**   For cost efficiency, and based on the strong performance of the Fast algorithm in Section 3.4, as well as our observation that it produces similar results, the cost list $C$ includes only a selected subset of possible cost values.

In particular, for the three drug review instances, we use $C = [80, 70, 60, 50, 40, 36, 32, 28, 24, 20, 16, 12, 10, 8, 6, 4, 2]$, while for bowlinggreen which has a different word budget per agent, we use $C = [80, 60, 40, 36, 32, 28, 24, 20, 16, 12, 8, 4]$.

**Approval Levels**   We use $\ell = [5.5, 5, 4.5, 4, 3.5, 3, 2, 1, 0]$ for each of the instances.

**Minimum Statement Lengths**   To ensure that meaningful statements get added to the slate, we impose a minimum statement length of 10 for the drugs dataset and of 8 for the bowlinggreen dataset. The minimum statement length is waived once we reach $\ell = 0$.

### D.4. Evaluation Chain-of-Thought Discriminative Queries

We use the following query in our evaluation of the generated slates to compute reliable utility values for agent, statement pairs:

**System Prompt**

```
You will be provided with survey responses from a user, and a separate statement.
Your task is to determine the extent to which the user would agree with that
statement, on a scale from 1 to 6. Your response should be JSON containing your
responses for each step in reasoning.
```

**Prompt**

```
**User description:**
<insert user information>

**Statement:**
<statement>

**Instructions:**

To determine the extent to which the statement fully summarizes the user's
opinion and reasoning behind it, follow these steps. Be very concise when
addressing each step.

Step 1. Summarize the user's opinions and reasoning on the topic. Include a few
concrete examples in your summary.

Step 2. Explain which aspects of the user's opinion and reasoning the statement
fails to touch on.

Step 3. Explain which aspects of the user's opinion and reasoning the statement
actively contradicts.

Step 4. Weigh all of your considerations thoughtfully, to determine overall how
much the statement summarizes the user's opinion and reasoning. Then select from
one of the following 6 choices.
{
6: "the complete details of the user's opinion and reasoning are captured by the
statement",
5: "almost all of the details of the user's opinion and reasoning are captured by
the statement, with very minor contradictions",
4: "around two-thirds of the details of the user's opinion and reasoning are
captured by the statement, with few contradictions",
3: "a majority of the details of the user's opinion and reasoning are captured by
the statement, with some omissions / contradictions",
2: "the user would at best slightly agree with the statement, because the
statement only partially captures their opinion and reasoning, or is missing
something",
1: "the statement is either heavily incomplete, or contradicts / is orthogonal to
the user's opinion and reasoning",
}

**Output instructions:**

Respond in JSON as follows:
{{
"step1" : <your response to step 1>,
"step2" : <your response to step 2>,
"step3" : <your response to step 3>,
"step4" : <your response to step 4>,
"score" : <your score, a number between 1 and 6>
}}
```

### D.5. Verification of Discriminative Queries

In this section, we validate the two discriminative queries used in our experiments: first, the discriminative query used in PROSE, and second, the chain-of-thought discriminative query used in the evaluation.

For this validation experiment, we use publicly available up- and downvote data from the Polis *BowlingGreen* dataset. (To ensure a sufficiently large sample size, we use the full dataset consisting of 2031 users, as opposed to the curated dataset of 41 users used in the rest of our experiments.) For each of the 51 users who placed at least 5 upvotes and 5 downvotes on comments they did not author, we select 5 upvotes and 5 downvotes uniformly at random. For each user-vote pair, we use a discriminative query to estimate how much the user approves the statement they voted on.

For both the discriminative query used in PROSE, and the chain-of-thought discriminative query, we observe that 84% (CI: 72.5% – 92.1%) of users have higher mean discriminative query scores for upvoted statements than downvoted statements. Thus, both queries can reliably predict user voting behavior from a user's written beliefs.

To verify the independence of the queries, for each discriminative query, we construct a vector of length 51, indexed by the users. Each user's entry is the difference between the mean discriminative query output on upvoted statements with the mean discriminative query output on downvoted statements. The Pearson correlation coefficient between these two vectors is 0.13—relatively low considering the high accuracy of both queries—demonstrating that the two discriminative queries are relatively independent, not just in their implementation, but also in their behavior.[15]

## E. Experiments: Additional Details

### E.1. Datasets

For the birth control dataset, we take all reviews on "Ethinyl estradiol / norethindrone" from the UCI ML Drug Review dataset (Gräßer et al., 2018). With the help of an LLM, we discard all reviews that speak about a specific brand. Out of the remaining 1275 reviews, we keep the reviews whose length is between the median and 75th percentile. For the uniform dataset, we sample 8 reviews with rating $x$ for each $x \in [1, 10]$. For the imbalanced dataset, we sample $20/10/20/10/20$ reviews with rating $1/2/5/9/10$.

For the obesity dataset, we take all reviews on "Contrave" and apply the same procedure as above, except that we keep all reviews whose length is between the 25th and 75th percentile (as there are fewer reviews for this drug).

### E.2. Baselines

#### E.2.1. CONTEXTLESS ZERO-SHOT

We query the LLM with the following prompt:

**System Prompt**

```
Your task is to write a *proportional opinion slate* on a particular topic while
staying within a word budget. A proportional opinion slate is a collection of
opinions that, taken together, give an overview of the general population's
opinions on that topic. Morover, the lengths of the opinions should correspond to
the relative proportion of people that hold that opinion (hence "proportional"
opinion slate).

**Stylized example:** Suppose 70% of people believe salads are the best dinner,
20% of people believe burgers are the best dinner, and 10% of people believe soup
is the best dinner, and the word budget is 50 words. Then, a proportional opinion
slate might look like this:
```

---

[15]The inter-rater reliability score (via Cohen's kappa) for the two implementations is 0.41 (fair to moderate agreement). For 39/51 users (76%), both implementations return higher mean scores on upvoted statements than downvoted statements; for 8/51 users (16%), one implementation returns higher mean scores and the other lower mean scores on upvoted statements (each implementation is correct 4/8 times).

```
- Salads are the best dinner option by far. They are healthy and tasty. They
fulfill all major dietary requirements so anybody can have them. They are also
flexible, since many different ingredients can be substituted.
- Burgers make the best dinner because they're tasty and filling.
- Soup makes the best dinner.

**Writing guidelines:** Each statement should be a short, strong opinion that
reflects a single, clear stance reflecting a population segment's thoughts. The
opinion should sound personal, direct, and conversational, as if written by
someone expressing their own thoughts. Avoid summary-style language or listing
multiple viewpoints. Finally, do not list word counts next to your statements --
just put the statements and nothing else.

The topic: {topic}
The word budget: {word_budget} words. Do not write more than {word_budget} words
under any circumstances!
```

**Prompt**

```
Write your proportional summary below. Write each statement on a new line, and
use "- " as bullet points. Respect the word limit!
```

### E.2.2. ZERO-SHOT

We use the same system prompt as for contextless zero-shot with the following prompt:

```
To improve the accuracy of your proportional opinion slate, below provided are
representative opinions that people have on the topic. You should aim to
construct a proportional opinion slate that matches the distribution of these
opinions.

{user_opinions_list_str}

Write your proportional summary below. Write each statement on a new line, and
use "- " as bullet points. Respect the word limit!
```

### E.2.3. CLUSTERING

For the clustering baseline, we start by computing an embedding of the description of each agent using OpenAI's `embedding-3-large`. Subsequently, we apply PCA compressing the embeddings into five dimensions. Finally, we compute a clustering using Affinity Propagation. For each identified cluster of $\ell$ agents, we generate a statement for the cluster using the prompt from Appendix D.2.3 with a word limit of $\frac{\ell B}{n}$.

### E.3. Generated Slates

In this section, we present the slates generated by all methods we consider. Note that the slate constructed by PROSE is typically a bit fragmented with multiple statements covering similar opinions One reason for this is the inner workings of the algorithm. The following pattern regularly emerges across instances:

We start with a high approval level. At this stage, a consensus statement is generated that receives strong—but not unanimous—support among users with a similar opinion. The statement does not receive the required approvals to get added at this high level (but would get added at a slightly lower level).

As the budget decreases, a shorter statement targeted to a smaller subset of the user group with similar opinions is created, which is evaluated highly by those users and is thus added to the slate. (The success of these short statements is in part due

to the very limited data we have available for each user; many users have only a few core points, which can be adequately captured within a succinct statement).

After one or two such short statements have been added, the approval level is decreased. At this lower level, the previously generated "long" consensus statement would have sufficient support to be added in case all agents from the dataset were still present. However, at this point, some users that would be covered by the "long" statement have already been removed due to the prior selection of a short statement. As a result, despite the lowered approval threshold, the original "long" consensus statement still does not have sufficient support to be added. This cycle then repeats itself across iterations and datasets.

Notably, this pattern does lead to agents having a high utility for their matched statement under our discriminative query.

### E.3.1. BIRTH CONTROL (UNIFORM)

**PROSE**

- This birth control is a nightmare—severe side effects outweigh any benefits. Avoid it at all costs.

- This birth control is worth sticking with; initial side effects fade, and benefits are undeniable!

- This pill wreaks havoc on your body and emotions.

- This birth control wreaks havoc on emotions, skin, and weight.

- These pills cause more harm than good; I'm switching.

- This birth control is misery in a pill, avoid it.

- This pill is life-changing—clear skin, lighter periods, and boosted confidence outweigh mild, manageable side effects!

- This pill works well after initial side effects settle down.

- These pills mess with mood, weight, and overall well-being.

- This birth control works, but the side effects can be frustrating!

- This pill works well after initial side effects subside.

- This birth control causes unpredictable side effects; I'd think twice before committing to it long-term.

- This pill is not worth it; side effects outweigh any benefits.

- Disappointing.

**PROSE-UnitCost**

- This pill is an absolute nightmare. It wrecks your body with relentless side effects—crippling cramps, constant bleeding, unbearable headaches, mood swings that spiral into depression, and even hair loss. It doesn't fix acne, it piles on weight, and the emotional toll is devastating. Sure, it might prevent pregnancy, but at what cost? No one should have to endure this kind of misery for birth control. I wouldn't recommend it to anyone—ever.

- This birth control pill is absolutely worth sticking with despite the initial side effects. Yes, the first few weeks can be rough—nausea, mood swings, spotting, or even weight changes—but once your body adjusts, it's a game-changer. It regulates periods, clears up skin, and doesn't cause major issues like bloating or severe mood swings for most people. Taking it at the same time every day, preferably at night, seems to minimize side effects significantly. Honestly, the benefits far outweigh the temporary discomfort, and for many, it's the best option they've tried. Just be patient and give it time—it's worth it.

- This birth control is a rollercoaster of side effects that makes me question if it's worth it. Sure, it prevents pregnancy, but at what cost? Between the weight gain, mood swings, acne flare-ups, constant spotting, and a complete nosedive in sex drive, it feels like trading one problem for a dozen others. Even the "positives" like bigger breasts or lighter periods come with strings attached—tenderness, irregularity, or worse. It's exhausting to deal with unpredictable bleeding, emotional chaos, and physical discomfort, all while hoping my body will "adjust." Honestly, this pill feels more like a gamble than a solution, and I'm not sure it's one I want to keep taking.

- This birth control is a mixed bag of extremes—either it works wonders or wreaks havoc. While it's effective for preventing pregnancy and managing issues like ovarian cysts, the side effects are all over the place: mood swings that ruin relationships, unpredictable bleeding, migraines, weight changes, and even terrifying ER visits for seizures or vision loss. Sure, some people see improvements in acne or lighter periods, but the trade-offs—like constant spotting, emotional instability, or debilitating cramps—make it feel like a gamble. Honestly, it's hard to trust something that can either fix your life or completely derail it.

- This pill is a mixed bag—while it works wonders for some, regulating periods, clearing skin, and even eliminating periods entirely, the side effects are no joke. From weight gain, mood swings, and breast tenderness to devastating consequences like blood clots, it's clear this isn't a one-size-fits-all solution. Personally, I'd be cautious and prioritize thorough discussions with a doctor before committing to it. The potential risks are too significant to ignore, no matter how convenient the benefits might seem.

**Clustering**

- Deadly drug.

- This medication ruined my health completely.

- This pill is a nightmare—weight gain, breakouts, and awful side effects!

- This birth control has been a nightmare for me—constant bleeding, horrible acne, mood swings, and unbearable cramps. It's just not worth the pain and frustration.

- This birth control wreaked havoc on my body—weight gain, no libido, mood swings—never again.

- This pill is a nightmare. The severe cramping, irregular bleeding, mood swings, and other distressing effects make continuing it feel unbearable and unhealthy.

- This pill is life-changing when used correctly. Stick through the initial side effects, take it at the same time daily, and you'll see amazing results!

- Honestly, this pill is a game-changer—clear skin, no weight gain, no wild side effects, and barely any periods. Totally worth every penny!

- This pill can be tough at first, but sticking with it is worth the positive changes later!

- Adjusting takes time, but overall it's effective!

**Zero-Shot**

- This medication was devastating for me. It caused severe side effects like persistent acne, mood swings, weight gain, and intense cramps. I would never recommend it to anyone.

- I experienced horrible nausea, moodiness, and spotting in the early months, but things improved over time. It helped lighten and regulate my periods, and now I'm mostly fine on it.

- I've experienced some discomfort, but overall it's effective at preventing pregnancy and hasn't been unbearable. Side effects vary, but they're manageable for me.

- This pill has been great! My periods are lighter, my acne has cleared, and I don't deal with cramps anymore. I highly recommend it!

**Contextless Zero-Shot**

- I've had no major side effects, and it's been a life-changing solution for me.

- It's mostly fine, but I do experience mood swings and mild nausea occasionally. It's effective overall, but the side effects can be a bit frustrating.

- I don't like using this medication. It caused me severe headaches and weight gain, so I had to stop. It just did not work for me.

- Honestly, it didn't change much for me. My cycles are still irregular, and I'm not sure if it's worth continuing.

- I've had a great experience with this medication. It cleared up my acne and regulated my hormones, so I feel much better overall.

- The side effects like dizziness and spotting made it unbearable for me. I had to switch to something else.

E.3.2. BIRTH CONTROL (IMBALANCED)

**PROSE**

- This pill is a nightmare. It wrecked my skin, my emotions, and my body. The side effects are unbearable—please, stay far away from it.

- This pill is unpredictable and risky; while it helps some, the severe side effects make it not worth the gamble.

- This birth control is life-changing—light periods, minimal side effects, and incredible reliability. Absolutely love it!

- This pill causes too many side effects to be worth it.

- Be patient; benefits outweigh initial side effects over time.

- This birth control is effective with minimal, manageable side effects.

- This birth control causes too many unbearable side effects to be worth it.

- This pill works great with minimal, manageable side effects!

- Birth control shouldn't rob you of your sex drive and emotional stability.

- This pill works well with manageable side effects—definitely worth trying!

- Birth control works, but side effects can be frustratingly unpredictable.

- This pill works, but side effects can be unpredictable and concerning.

**PROSE-UnitCost**

- This birth control might prevent pregnancy, but the side effects are absolutely not worth it. From relentless nausea, mood swings, and acne to weight gain, emotional instability, and even painful sex, it feels like trading one problem for a dozen others. It's exhausting to deal with irregular periods, constant spotting, and feeling like your body is completely out of sync. Honestly, no one should have to endure this much just for contraception. I'm done with it.

- This birth control pill is a game-changer. It's incredibly effective, lightens periods to the point of barely noticing them, and for many, clears up skin and eliminates cramps. Sure, the first couple of months can be rough with side effects like mood swings, nausea, or acne, but once your body adjusts, it's smooth sailing. The key is consistency—taking it at the same time every day makes all the difference. It's not perfect for everyone, but for those it works for, it's life-changing. I'd recommend it in a heartbeat.

- This birth control is a gamble I wouldn't take again. Sure, it prevents pregnancy, but at what cost? Between the nausea, weight gain, mood swings, loss of libido, and even life-threatening risks like blood clots and seizures, it feels like playing Russian roulette with your health. Some people might love it, but for me, the side effects are just too unpredictable and severe to justify sticking with it. No thanks.

- This pill is a mixed bag of extremes—while it's effective at preventing pregnancy and can clear up acne or regulate periods for some, the side effects are no joke. From nausea, mood swings, and breast tenderness to terrifying experiences like vision loss, seizures, or hair falling out, it's clear this isn't a one-size-fits-all solution. Personally, I'd be hesitant to recommend it unless you're prepared to endure a rollercoaster of side effects while hoping your body adjusts. It works, but at what cost?

- This birth control is a mixed bag—while it works effectively for many and even improves things like acne, lighter periods, or sex drive, the side effects can range from mild nausea and breast tenderness to terrifying health scares like seizures or even fatal blood clots. Honestly, it's a gamble with your body, and while some swear by it, others have paid the ultimate price. Personally, I'd be too scared to risk it.

## Clustering

- This drug is dangerous.

- This pill causes unbearable cramps and prolonged, unpredictable bleeding.

- Worst medication ever; caused unbearable side effects.

- This pill is dangerous, avoid it!

- This birth control has caused relentless acne, mood swings, and physical discomfort—definitely not worth the misery.

- This pill causes more harm than good.

- This birth control is a nightmare. The side effects are unbearable – from weight gain, mood swings, and nausea to painful headaches and zero sex drive. It's simply not worth the toll it takes on your body and mind.

- Birth control ruined my moods, skin, and sex drive—definitely not worth it!

- This pill works great once your body adjusts. Sure, side effects happen, but the benefits outweigh them!

- I absolutely love this birth control! It cleared my skin, made my periods super light or nonexistent, and stabilized my mood. No major side effects for me—definitely worth the cost!

- This pill significantly reduces cramps and heavy periods effectively!

## Contextless Zero-Shot

- My experience with Ethinyl estradiol / norethindrone has been overwhelmingly positive. It regulated my periods, cleared my acne, and gave me a sense of control over my body. I haven't experienced any major side effects, and I feel confident using this medication as part of my routine.

- It works well enough for me, but I do experience mood swings and some nausea occasionally. While it's effective for birth control and cycle regulation, these side effects make it less than perfect for me.

- The side effects have been too much for me to handle—constant bloating, headaches, and emotional ups and downs. I stopped taking it because I didn't feel like myself anymore.

- It caused me significant issues like weight gain and unpredictable spotting. I've decided to try something else because this medication just isn't for me.

- I didn't notice much of an effect at all, positive or negative—it just wasn't the right choice for me.

**Zero-Shot**

- This medication has been an absolute nightmare for me. My anxiety, mood swings, and depression have worsened dramatically since starting it. My periods have become irregular and extremely painful, and the side effects have made me feel like I'm losing control of myself. I cannot recommend this to anyone.

- I experienced terrible weight gain and breakouts after taking this pill. Despite following a strict diet and exercise plan, I haven't been able to shed the weight even after stopping the pill. It's completely thrown my body out of balance and I regret trying it.

- My experience has been awful. I've had severe nausea, constant bleeding, abdominal cramps, and intense headaches. My mental health has also been affected, making me irritable and emotional. I gave this pill a chance, but I wouldn't wish it upon anyone else.

- The side effects of this pill made me feel terrible—headaches, nausea, bloating, and zero energy. My periods became unpredictable and heavy. While it may work for some, it just didn't work for my body.

- This birth control has been effective in preventing pregnancy, although I've experienced minor side effects like nausea and acne. It's been manageable overall and did help regulate my period after the first few months.

- I've had a great experience with this medication. It cleared up my acne, regulated my periods, and made my cycles lighter and more predictable. I'm happy with the results and would recommend it to others trying to manage similar issues.

- This pill has been a game-changer for me. Minimal side effects and no periods, which I love. I've had clearer skin, steady moods, and it's been a reliable form of contraception. Definitely worth sticking with.

### E.3.3. OBESITY

**PROSE**

- This pill is pure misery—nausea, dizziness, and constant sickness!

- This medication curbs cravings, boosts energy, and transforms eating habits!

- Weight loss isn't worth these awful side effects and struggles.

- This medication is a game-changer! It curbs cravings, boosts control over eating, and delivers real weight loss results despite minor side effects. Totally worth it for serious lifestyle changes!

- This pill powerfully curbs hunger, but side effects can hit hard.

- This drug's side effects are unbearable and not worth it.

- This pill kills cravings, but effort and lifestyle changes matter!

- This pill works for weight loss, but the nausea and side effects make it hard to continue.

- This medication curbs cravings but demands effort and lifestyle changes.

- This journey is tough, but hope and persistence will bring results.

- This drug causes unsettling side effects; take it cautiously.

- This medication is frustratingly ineffective for weight loss, with minimal results and unpleasant side effects.

- Disappointed hope.

**PROSE-UnitCost**

- If you're serious about changing your life, this medication can be a game-changer, but it's not a magic fix—you have to put in the work. The appetite suppression is real, cravings diminish, and for many, the weight starts to drop, but side effects like nausea, fatigue, dry mouth, and headaches can make the first weeks rough. If you're ready to push through and commit to healthier choices, this can be the tool to help you get there. Just don't expect results without effort.

- This medication is a rollercoaster—sure, it curbs appetite and helps with cravings, but the side effects are brutal. From extreme nausea, dizziness, and exhaustion to feeling completely out of it, it's like trading one struggle for another. Honestly, it's hard to justify sticking with something that makes you feel so awful, even if the scale moves a little. If your body can handle it, great, but for many, it's just not worth the misery.

- This pill can be a game-changer if you're ready to push through the initial side effects like nausea, headaches, or fatigue. It's not a magic solution, but it genuinely curbs cravings, forces better food choices, and can lead to significant weight loss if you stick with it. However, it's not for everyone—some people can't tolerate the side effects, and results vary. If you're serious about change and can handle the discomfort, it's worth trying, but don't expect miracles without effort.

- This medication is a mixed bag of frustration and faint hope. It's riddled with side effects—anxiety, nausea, constipation, exhaustion, and even mental fog—but delivers little to no weight loss for most. Sure, it curbs hunger for some, but cravings persist, and the scale barely budges. A few users see minor benefits like pain relief or reduced snacking, but the trade-off is steep. Honestly, it feels like a gamble with your body for results that just don't justify the misery. I wouldn't recommend it.

- This medication is a complete mixed bag—while it can suppress appetite and lead to weight loss for some, the side effects are absolutely brutal and often outweigh the benefits. From crippling headaches, dizziness, and insomnia to anxiety, confusion, and even worsened heart conditions, it feels like trading one problem for a dozen others. Sure, a few people see results, but the majority seem stuck battling constant discomfort, unpredictable side effects, and minimal progress. Honestly, it's not worth the gamble on your health.

**Clustering**

- Stay away; side effects are unbearable!

- Useless and dangerous—never taking it!

- This pill causes too many side effects; it's not worth the discomfort.

- This medication's side effects are intolerable.

- The side effects are unbearable, making this pill not worth taking.

- Nausea and dizziness are tough, but this is worth trying.

- This med helps curb cravings and appetite, but side effects like dry mouth, tiredness, or nausea are common early on.

- This medication seems to mess with my mind and body too much for the minor benefits it provides.

- This pill works well to curb appetite and support weight loss, but dedication and lifestyle changes are essential.

- This medication isn't worth the cost or side effects for inconsistent weight loss results.

- This journey proves commitment and effort bring amazing results.

- This medication works but the side effects can be tough.

- This medication works for weight loss, but the side effects can be overwhelming and frustrating.

**Zero-Shot**

- I had a terrible experience. The nausea, headaches, and dizziness were unbearable. I couldn't eat, felt constantly sick, and had to stop within days. I wouldn't recommend this medication to anyone.

- I've been on Contrave for weeks and haven't lost weight. The side effects, like nausea and tiredness, make it hard to continue. I'm frustrated that it's not working as I hoped.

- This medication suppressed my appetite, but the side effects like dry mouth, dizziness, and digestive issues made it hard to tolerate. I did lose some weight, but I'm unsure if I'll keep taking it.

- Contrave is helping me lose weight, and I've noticed fewer cravings. The side effects were manageable and improved over time. I feel optimistic and healthier.

**Contextless Zero-Shot**

- Contrave has been a game-changer for me. It significantly reduced my cravings and helped me control my appetite. I've lost weight steadily and feel like I'm finally in control of my eating habits.

- I noticed some initial side effects, like nausea and dizziness, but they went away after the first couple of weeks. It's been helpful for my weight loss journey overall.

- Honestly, Contrave didn't work for me. I stuck with it for a couple of months, but the side effects were too much to handle, and I barely saw any weight loss.

- It's okay, but I feel like the results are slower than I expected. It helps curb my appetite a bit, but I still need to work hard with diet and exercise.

- Contrave caused too many issues for me. I was constantly nauseous, couldn't focus, and felt worse than before I started it. I had to stop taking it altogether.

E.3.4. BOWLING GREEN

**PROSE**

- Refugees enrich communities and deserve warm welcomes.

- Warren County deserves countywide high-speed internet as a basic public utility!

- Traffic laws need strict enforcement for safer roads.

- Traffic needs fixing before any new developments.

- Every family deserves true school choice access!

- Animal cruelty laws must be strict and enforced.

- Immigrants need better integration and support systems.

- Sex ed must be comprehensive and medically accurate.

- Traffic flow improvements are urgently needed; congestion makes driving unsafe and frustrating.

- Traffic flow improvements and safer intersections are urgently needed in Bowling Green.

- We absolutely need faster, affordable internet expanded countywide to underserved areas.

- BG needs traffic cameras to improve safety now.

- School district boundaries urgently need major reform.

- Internet options must expand for residential access.

- Bowling Green desperately needs better traffic solutions.

- Bowling Green needs fair access to education.

- Bowling Green desperately needs more pet resources!

- Schools must address bullying immediately and effectively.

- Fund arts education now.

## PROSE-UnitCost

- Bowling Green desperately needs to fix its traffic chaos—limit unnecessary lights, add turn lanes, block left turns on busy roads, and install traffic cameras to enforce order. It's ridiculous how poorly planned and congested our roads are, especially during rush hours. And while we're at it, let's hold BGMU accountable for their rates, expand internet infrastructure, and finally give this city an ice rink and a proper bypass. These are basic improvements that would make life here so much better.

- Every child deserves access to quality education, whether through better-funded arts programs, equitable school choice, or addressing overcrowding in Warren County schools. It's time to prioritize students' needs over outdated district lines and ensure every family, regardless of income or location, has the opportunity to choose the best education for their kids.

- Stop wasting taxpayer money on frivolous projects like fountains and start addressing real community needs—fix the traffic nightmare, enforce parking laws, and invest in education and youth programs. Overdevelopment of rentals is killing our schools, and we need a stronger tax base to support them. Also, animal welfare laws must be enforced, and pet owners need to be held accountable. Build a homeless park with tiny homes to give people dignity, and make public spaces safer and more accessible for everyone, especially the elderly and disabled. Prioritize people over cars, enforce parking regulations, and create a city that works for its residents, not just developers or convenience. Enough is enough—it's time for real change!

- BGMU needs to stop hiding behind vague "averages" and actually deliver affordable, reliable internet to all residents—this is 2023, and people in surrounding counties have better options! Also, Bowling Green schools are failing students by ignoring bullying and refusing to implement real, comprehensive sex education. It's time to stop pretending these issues aren't affecting our community. And don't even get me started on the traffic mess at Shive Lane or the lack of a proper bypass—fix it already!

- Bowling Green thrives because of its diverse refugee and immigrant communities, but the city desperately needs to hold landlords accountable for the overpriced, crumbling rentals they offer. It's time to demand better housing options, including pet-friendly spaces, and ensure new developments don't outpace the infrastructure we rely on. Let's fix the traffic mess on Fairview with a proper roundabout and stop approving projects that strain our roads. This city can do better, and it starts with prioritizing its people and their quality of life.

## Clustering

- Fairview Ave desperately needs a traffic circle at Kereiakes Park to fix the chaotic flow and reduce reckless driving through red lights.

- Internet should be a publicly managed utility, with residential fiber access prioritized. BGMU must lead, ensuring equitable, high-speed internet for all Warren County residents.

- Parking laws need strict enforcement—cars clogging streets or ignoring rules create chaos and danger. Police should actively ticket violators and restore order.

- Comprehensive, accurate sex education is essential to combat teen pregnancy and STDs.

- The school district boundaries and overcrowding issues in Warren County need serious attention. Redistricting and building new schools could better balance student numbers and meet the demands of a growing population.

- Bowling Green desperately needs to address its traffic chaos; adding turn lanes, blocking left turns on busy roads, and improving traffic flow should be top priorities for the city!

- Landlords must ensure safe, affordable rentals; no more overpriced, run-down units!

- Refugees are welcome!

## Zero-Shot

- We desperately need faster, more reliable internet across Bowling Green and Warren County, especially in underserved areas like Alvaton.

- Traffic in Bowling Green is a nightmare. The Bypass doesn't function like a real bypass, and Fairview Avenue near Kereiakes Park needs a roundabout to improve flow.

- Schools at all levels need daily physical activity to combat childhood obesity. It's not optional—it's a necessity for our kids.

- Refugees and immigrants enrich our city. We should implement programs to integrate them better into our community and celebrate their contributions.

- Build a homeless park with tiny homes and a central donation center. Our city has to care for its most vulnerable.

- Roads and infrastructure should improve alongside new developments. We can't keep approving projects without a plan for traffic and utilities.

- More needs to be done to protect animals. Bowling Green should ban chaining pets and enforce spay/neuter programs through fines.

- More affordable, accessible transportation for seniors and disabled residents would make life better for everyone.

- Enforcing traffic laws like speeding, running red lights, and expired tags would improve safety and bring in more revenue for the city.

- We need more housing options for renters that allow pets, and landlords should be held accountable to maintain their properties.

## Contextless Zero-Shot

- Bowling Green desperately needs better road infrastructure. The traffic is unbearable during peak hours, and the road conditions are poor. Expanding roads and improving maintenance should be a top priority.

- We need more activities and attractions for families and young people. Investing in parks, community centers, and events would help bring the community together. Bowling Green has potential, but it feels stagnant.

- Affordable housing should be a focus. Prices are climbing, and it's becoming harder for young families and low-income residents to stay here. More housing developments with reasonable prices are necessary.

- Public transportation here is almost nonexistent. We need better bus routes and other transport options for those who don't drive, especially for seniors and students.

- The city could benefit from attracting more diverse businesses. Local jobs are great, but more opportunities in technology or creative industries would bring growth.

- Investing in education and local schools is key. Schools need better funding, resources, and attention.

- Bowling Green is fine as it is. People complain too much.

# F. Additional Discussion

We featured in the discussion in the main body that more powerful queries would allow us to implement classic PB algorithms. Consider the setting with approval preferences (which corresponds to our setting when setting $r = 2$). We claim that we could simulate sequential-Phragmén given access to the discriminative query and the following generative query: Let $D = (d)_{i \in N}$ be a collection containing each agent's $i$'s current debt. Agents can spend money on a statement but this will increase their debt accordingly. Given $D$, the generative query returns the statement that can be paid for by its approvers and minimizes the maximum debt of an approver. This will correspond to: $\operatorname{argmin}_{\alpha \in \mathcal{U}} \frac{c(\alpha) + \sum_{i \in N_\alpha} d_i}{|N_\alpha|}$, where $N_\alpha$ is the set of approvers of statement $\alpha$. This query is sufficient to implement the discrete formulation of sequential-Phragmén, e.g., as described by Rey & Maly (2023, Definition 4).

Additionally, more complex queries would also allow us to aim for additional axiomatic guarantees. For instance, EJR+ up to any project for cost utilities as defined by Brill & Peters (2023, Definition 16) could be achieved given access to the following generative query: Let $\ell, x \in \mathbb{N}$ and $S$ be a group of agents. Generate the statement $\alpha$ that satisfies $\frac{\ell + c(\alpha)}{|\{i \in S \mid u_i(\alpha) = 1\}|} \leq x$ and maximizes $|\{i \in S \mid u_i(\alpha) = 1\}|$ if existent.

