# OpenReview forum: "Generative Social Choice: The Next Generation"
_ICML.cc/2025/Conference — ICML 2025 oral_

### Official Review · Reviewer_PwBp · 2025-02-18

**Overall Recommendation:** 5

**Summary:**

The paper investigates the scope of generative social choice, aiming to generate a slate of statements (usually from a large set such as textual information) representing the voters. Queries (usually implemented by LLMs) can be made on agent utilities and on generating a good statement among a group of agents. The problem is naturally related to traditional committee voting and large language models, which have strong generative power.

The paper extends previous literature by taking the overall budget and the inaccurate queries into consideration. The paper proposes a new algorithm that generates the slate (and assigns agents to statements). Theoretically, the paper gives worst-case guarantees on the performance of their algorithm reaching the balanced justified representation axiom parameterized by the errors in the queries. They also somewhat show that the guarantee is close to the optimum. Secondly, they show in synthetic experiments that the performance converges way faster than the theoretical guarantee under the same setting. Finally, they implement the generative social choice system PROSE by GPT-4o and test it on real-world data. It outperforms all the benchmarks in both keeping the BJR axiom and maintaining a high utility of the generated slates among the agents.

## After Rebuttal

The author(s) address my questions well. I am happy to maintain the current score.

**Claims And Evidence:**

Yes. All the claims are supported by theories or experimental results.

**Essential References Not Discussed:**

No.

**Experimental Designs Or Analyses:**

I am not very familiar with examining experiments, but in general, especially the validation experiments, looks god to me.

**Methods And Evaluation Criteria:**

In general, yes. This paper uses LLM to evaluate the average utility of the outcome slate in real-world experiments. The evaluation method is seemingly similar to the implementation of the DISC query (utility query) in their algorithm, but they run validation experiments to show that they are likely independent.

**Other Comments Or Suggestions:**

Typo:
Line 2546-252: I suppose $\zeta$ and $\alpha^*$ refer to the same statement?

**Other Strengths And Weaknesses:**

Strengths: I think this is in general a very good paper. It studies a topic full of potential (generative social choice) and gives a rigorous and practical solution. They aim to address the budget constraint and the inaccuracy of the queries (usually comes from the unstableness of LLMs) and solve the problem with a newly designed, theoretically guaranteed algorithm. The algorithm and the theories are not direct extensions of previous literature. Impossibility results are also given to illustrate the (nearly) closedness of the worst-case bound. Moreover, they implement the algorithm in practice and demonstrate its capability on real-world data. The paper significantly removes the assumptions and broadens the application of generative social choice, making this AI-augmented voting schema more promising. The presentation is also good, with most of the important assumptions or designs explained (such as BJR) or validated via experiments.

Weaknesses:
I don't have major complaints about the paper, yet some points are worth improvements

1. Proof for Theorem 3.1 is a bit dense for now. As this is the only proof in the main paper, I suggest that give a clear explanation of every important step to convey your key theoretical techniques.

2. The paper can benefit from a more clear discussion between itself and [Fish et al, 2024] on, for example, why its result is not a natural extension of the existing work.

3. The impossibility results consist of multiple theorems in which some parameters are fixed to 0 or 1 (the exact version). Are there more general results?

4. The implementation of the GEN queries in PROSE seems different from those in the theories. Instead of finding a statement with max support in a given subset, it seems to cluster agents and find statements for clusters. How do you justify this?

**Questions For Authors:**

1. How do you justify your implementation of GEN queries in PROSE? Does your implementation still follow your theory?

2. Are there more general impossibility results where parameters are not fixed on special values?

3. You assume the cost is only on the output. Given the implementation requires a considerable number of LLM calls, do you think it is more reasonable to also take query costs into consideration?

4. Your theoretical results only consider the satisfaction of BJR, while your experiments have positive results on the average utility of the outcome slate. Does your theory also show something about the utility?

5. The error parameter $\mu$ only works on comparing the query output with other potential statements. I think this needs a justification. Why can the query output a statement with an accurate cost, but the comparison is made under $\mu$ error?

6. Is there any runtime evaluation on PROSE (and probably other benchmarks) and between different procedures in PROSE?

**Relation To Broader Scientific Literature:**

The paper follows a very newly-emerged yet heated topic of generative social choice [Fish et al, 2024], which has the potential to open an utterly new research direction in this area. This paper extends the idea to more practical settings, making the prospect more realistic. The paper is also closely related to the combination of LLM and social choice/multi-agent systems, bringing insights into AI-augmented voting and decision making.

**Theoretical Claims:**

I check Theorem 3.1 and browse Theorem 3.2. I think they are correct.

---

> ### Author Rebuttal · Authors · 2025-03-30
>
> We thank the reviewer for their feedback. We will improve the clarity of the proof of Theorem 3.1 and expand the discussion on its relation to Fish et al.; see also our response to Reviewer dWvn.
>
> > How do you justify your implementation of GEN queries in PROSE? Does your implementation still follow your theory?
>
> Our theoretical results are agnostic to the implementation of the generative query. In this sense, we view different query implementations as approximate methods for solving the optimization problem defined by Equation (1).
>
> In practice, we experimented with various implementations and ultimately adopted a two-step procedure. First, we identify agents $S' \subseteq S$ who are likely to approve the same statement at level $\ell$, using embeddings and clustering or nearest-neighbor techniques. Second, we prompt the LLM to generate a length-bounded statement that would be approved by all agents in $S'$. That is, we first identify a potential supporter group and then generate a statement intended to satisfy them. We adopted this approach because LLMs were unreliable at discovering such groups. Ultimately, when presented with a generative query, we always produce multiple candidate statements, from which we return the one with the highest number of supporters at level $\ell$.
>
> > Are there more general impossibility results where parameters are not fixed on special values?
>
> Let us consider Theorem 3.4 as an example. The result is stated under the assumption of no error in how statements are evaluated (i.e., $\beta = \delta = 0$). Conditioned on this, the theorem shows that no algorithm can guarantee $(p,\frac{1}{\mu}\frac{|W|}{|W|\gamma+1})$-cBJR for any constant $p\in \mathbb{N}_0$.
>
> The immediate implication of this is that such a guarantee is also impossible for any other value of $\beta$ and $\gamma$ (adding error will never help). One might ask whether Theorem 3.4 can be strengthened for non-zero $\beta$ or $\gamma$, but this seems unlikely: Increasing them intuitively only drives up $b$ in $(b,d)$-cBJR. However, Theorem 3.4. already shows that we cannot get a guarantee for any fixed $b$ even for $\beta=\gamma=0$. We will make this implication more explicit in the paper.
>
> > Do you think it is more reasonable to also take query costs into consideration?
>
> This is an interesting point, but somewhat orthogonal to the primary focus of this paper. Our goal is to ensure that the output slate proportionally reflects user opinions – a challenging task in itself. That said, we agree this raises a possible direction for future work: one could consider imposing proportionality not only on the slate but also on the cost of the generation process. For example, one might impose that each agent "deserves" that $1 is spent on trying to generate consensus statements they like.
>
> > Does your theory also show something about the utility?
>
> Yes and no. We do not provide formal guarantees on average utility across all instances. However, our BJR guarantees ensure that sufficiently large and cohesive agent groups are mapped to statements in the slate from which they derive a high utility. For highly homogeneous instances, this would imply a non-trivial average utility.
>
> > Why can the query output a statement with an accurate cost, but the comparison is made under $\mu$ error?
>
> The role of the parameter $0 \leq \mu \leq 1$ is best understood in the absence of other errors. In this case, Equation (1) reduces to:
>
> $$ \text{sup}(\alpha^*, S, \ell) \geq \max_{\alpha \in \mathcal{U} : c(\alpha) \leq \lceil \mu x \rceil} \text{sup}(\alpha, S, \ell) $$
>
> This means that the returned statement $\alpha^*$, even if it has cost $x$, must only be as well supported as any statement of cost up to $\mu x$.
>
> We introduced this parameter because LLMs often undershoot the provided word budget in practice. Intuitively, the model would internally only search for statements in a more conservative space (shorter than allowed). As a result, we can only expect to identify the best statement among those of length at most $z$ (for some $z\leq x$). This behavior is captured by $\mu$, which quantifies this budget undershooting.
>
> > Is there any runtime evaluation on PROSE (and probably other benchmarks) and between different procedures in PROSE?
>
> The runtime of PROSE is dominated by the response times of the LLM used in our query implementations. Our algorithm makes $\mathcal{O}(r \cdot n)$ generative queries and $\mathcal{O}(r \cdot n^2)$ discriminative queries. In our experiments, across the four datasets, PROSE used 9.6M–25.4M input and 53.5K–96.1K output tokens, with runtimes of 31–65 minutes on a single Intel i7-8565U CPU @ 1.80GHz. However, given the rapidly improving inference speeds of modern LLMs, we expect these runtimes to significantly decrease in the future.
>
> By contrast, PROSE-UnitCost required around five times fewer resources: between 2.1M and 4.4M input tokens and 15.4K to 20.2K output tokens, with runtimes between 7 and 12 minutes.

---

> > ### Comment · Reviewer_PwBp · 2025-04-03
> >
> > Thank you for the explanation. I am happy to maintain the current score.

---

### Official Review · Reviewer_dWvn · 2025-03-09

**Overall Recommendation:** 4

**Summary:**

The authors consider the problem of generating a set of statements that is representative of a collection of agent opinions on some topic, motivated by participatory budgeting. Extending an earlier definition of proportionality for such a setting, balanced justified representation (BJR), the authors introduce a version with costs and approximate proportionality, (b, d)-costBJR. The authors consider an algorithm that has access to two types of (approximate) queries, (1) discriminative queries, which give the utility of agent $i$ for statement $\alpha$; and (2) generative queries, which for a set of agents, a utility threshold $\ell$, and a cost $x$, finds the statement with cost at most $x$ that maximizes the number of agents in the set who have utility for the statement at least $\ell$. These types of queries were introduced in prior work, but the authors extend them to the approximate case. They provide an algorithm which approximates $(b, d)$-cBJR if given access to approximate discriminative and generative queries. They also prove lower bounds matching the utility approximation error and nearly matching the proportionality approximation error. The authors demonstrate their algorithm's guarantee and impossibility results in a synthetic data experiment. Then, they propose an LLM-based approach, which they call PROSE, to heuristically implementing generative and discriminative queries for their algorithm (i.e., DISC = ask GPT-4o to estimate agent utilities for a statement given some text describing their opinion, GEN = ask GPT-4o to write a statement approved by a set of agents). The authors apply PROSE to several semi-synthetic real-world datasets, using a chain-of-thought version of their DISC prompt to estimate agent utilities for generated statements. PROSE performs substantially better in terms of (GPT-estimated) agent utilities and BJR violations than LLM baselines using clustering and zero-shot generation.

### Update after rebuttal
Thanks for the very clear answers! Incorporating this info into the paper/appendix would be great. I'm happy to continue recommending acceptance.

**Claims And Evidence:**

The theorems are all supported by proofs in the main text or appendix, and the experiments do a good job of demonstrating the effectiveness of the proposed algorithm.

**Essential References Not Discussed:**

None which I am aware of.

**Experimental Designs Or Analyses:**

I have some moderate quibbles with the some details of the experiment, but given the difficulty of evaluating generated statements, I think the approach the authors took is reasonable. I would have liked a little more spelling out of experimental weaknesses in the text, as described below.

An important comparison that's missing: it seems like ti should be possible to run the algorithm from Fish et al. (2024) and compare it to DemocraticProcess when used in PROSE. Is this new algorithm that accounts for approximate queries a meaningful improvement when used with LLMs?

- There is clearly some self-confirmation bias in using GPT-4o to compute "true" utilities against which the proposed methods are evaluated as well as the DISC queries in the execution of PROSE. But the fact that the baselines are given the advantage of using the true CoT utilities is a nice touch that alleviates this concern. It would be good for the paper to more explicitly highlight this source of bias and the mitigation strategy used to strengthen the baselines by using CoT utilities. Currently this point is relegated to footnotes. Then again, without an actual experiment with human subjects, I don't see a better way of evaluating the utility of generated statements.

- Section D.5 of the appendix presents two seemingly contradictory claims, that (1) both the DISC query and the CoT utilities have high correlation with true thumbs up/down ratings and (2) the two utilities are very weakly correlated. I can see why the authors want both of these things to be true, as (1) supports the validity of GPT-4o utility estimates and (2) mitigates makes the earlier issue of self-confirmation bias, but there's also a tension between those claims. The measurement of correlation (line 1113-1117) is very strange and hard to interpret. Some more intuitive measures that I would have liked to see: the inter-rater reliability score, the fractions of agent-statement pairs where the difference in labels is 0 and 1, and most comprehensive, the full distribution of label discrepancies for each agent-statement pair.

**Methods And Evaluation Criteria:**

I think the methods and evaluation criteria are reasonable.

**Other Comments Or Suggestions:**

1. If the cost of a statement is its length, it seems quite odd to require that the fraction of agents assigned to statement $\alpha$ is proportional to the the length of $\alpha$. It seems more natural to have weighted statements. E.g., with 10 agents, 3 of whom dislike ice cream and 7 of whom like it, it feels odd to require a set of word-length-balanced set of statements like {"hate ice cream", "love love love love love ice cream"} rather than: {"I really like ice cream": 0.7, "I don't like ice cream": 0.3} (same word budget of 10 in both cases). In other words, I feel like the unit cost version of the model makes more sense, but with the addition that statements should be labeled with the proportion of agents who support them at some meaningful level $\ell$
2. lines 351-352: "PROSE also does not require any tuning of hyperparameters". This is technically true, but only because the approach uses a trillion-parameter black-box LLM with no guarantees for its responses to GEN and DISC queries. Generating statements uniformly at random also requires no hyperparameter tuning--this is obviously an unfair comparison, and I'm sure GPT-4o does a remarkably good job of generating statements based on agent text. But highlighting the lack of hyperparameters feels a bit disingenuous.
3. I would find 4.1 less objectionable if it highlighted both the strengths and limitations of using an LLM for DISC and GEN queries: yes, massive flexibility in agent preference format (free text, survey responses, ...) and cutting-edge text generation quality, but also no guarantees due to the nature of LLMs and the possibility of errors/hallucinations. I think the Impact Statement does a great job of acknowledging the limitations of using an LLM for statement generation and discrimination, but I would like to see some of these points incorporated earlier into the paper, when PROSE is introduced.
4. Minor: in lines 751-757, this algebra is correct, but I think it's preferable to avoid the appearance of "affirming the consequent" (i.e, the fallacy that x is true because x implies true). Verifying that this indeed establishes the desired inequality requires being sure every chain is an if-and-only-if rather than merely an implication. I would reorder the argument to arrive at the desired conclusion rather than start with it

**Other Strengths And Weaknesses:**

Overall, I think this is a very nice paper, but with a few areas to improve. To summarize what I've written in other sections:

Strengths:
- The paper is extremely well-written and clearly presented. Overall quality is great.
- The paper has all of the components I like to see in a combination theory/applied paper: upper and lower bounds and application to real data.
- The problem is very interesting and the results are strong

Weaknesses:
- I'm not convinced by balancing on word length of statements. This seems convenient but is a very odd choice for real-world applications.
- The lack of comparison to the algorithm of Fish et al. (2024) either merits solid justification or it should be added
- The text of the paper could do a better job highlighting the weaknesses of LLMs rather than only focusing on their advantages (and I say this as someone who thinks an LLM is the best tool for this job)

**Questions For Authors:**

1. for the drug review datasets, why artificially rebalance the dataset to make the score distribution uniform or highlight extreme and central ratings? It seems like the more natural thing to do based on the motivation for proportionally representative statements is to use the full distribution of reviews (or at least a representative random sample if the dataset is too large).
2. Out of curiosity, any reason why the budgets were 160 in the drug review experiments and 164 in the Bowling Green experiment? Not important to the results, just seems like an arbitrary inconsistency.

**Relation To Broader Scientific Literature:**

This work builds very explicitly on the work of Fish et al. (2024), which is even reflected in the title. It is a clear improvement in the handling of approximate queries and having an algorithm with approximate guarantees.

One important thing that seems to be missing is a discussion of how the DemocraticProcess algorithm is similar or dissimilar to the algorithm of Fish et al. introduced for exact DISC and GEN queries. How do the algorithms differ?

**Theoretical Claims:**

The definitions and claims are very clearly presented, and the proofs I checked are well-written and convincing (Thms 3.1 and 3.4), although I have not checked every detail.

---

> ### Author Rebuttal · Authors · 2025-03-30
>
> We thank the reviewer for their comments! We will revise the paper to include a detailed discussion of the limitations of our experimental setup and LLM-based query implementations. We will also add a dedicated section elaborating on the relationship to Fish et al. (2024).
>
> > for the drug review datasets, why artificially rebalance the dataset to make the score distribution uniform or highlight extreme and central ratings?
>
> The main reason for this design decision is that the original score distributions are quite degenerate, e.g., in the obesity dataset, over 70% of users give a score of 9 or 10. This skewness makes the summarization task quite simple, as nearly all users would support the same kind of statements.
>
> > Any reason why the budgets were 160 in the drug review experiments and 164 in the Bowling Green experiment?
>
> The budget in each case was chosen to be divisible by the number of agents (n = 80 for drugs and n = 41 for Bowling Green). This avoids rounding artifacts in the Clustering baseline.
>
> > The measurement of correlation (line 1113-1117) is very strange and hard to interpret. Some more intuitive measures that I would have liked to see: the inter-rater reliability score, the fractions of agent-statement pairs where the difference in labels is 0 and 1, and most comprehensive, the full distribution of label discrepancies for each agent-statement pair.
>
> The inter-rater reliability score (via Cohen's kappa) for the two implementations is 0.41. For 39/51 users (76%), both implementations return higher mean scores on upvoted statements than downvoted statements; for 8/51 users (16%), one implementation returns higher mean scores and the other lower mean scores on upvoted statements (each implementation is correct 4/8 times). Disaggregating by user and looking at all 1275=51\*5\*5 (agent, upvoted statement, downvoted statement) pairs, both queries agree on 53% of pairs, CoT is correct and PROSE is incorrect on 12% of pairs, PROSE is correct and CoT is incorrect on 13% such pairs, and both are incorrect on 21% such pairs.
>
> > *Differences in democratic processes compared to Fish et al. (2024) and missing experimental comparison*
>
> Our democratic process is related to the one of Fish et al. but differs in several aspects:
> 1. We account for statements having varying costs (e.g., word lengths).
> 2. Fish et al.’s generative query, given a set of agents $S$ and an integer $r$, returns the statement maximizing the $r$-th highest utility among users in $S$. In contrast, our query takes a utility threshold $\ell$ and returns a statement approved by the maximum number of agents in $S$ at level $\ell$.
> 3. Related to 2., our algorithm iteratively considers decreasing utility levels to decide which statement to add next, which is important for deriving our proportionality guarantees under approximate queries.
>
> As for the absence of a direct experimental comparison: unfortunately, the implementation by Fish et al. (2024) is not applicable to our datasets, as it requires highly structured user input (e.g., ratings of predefined statements and answers to survey questions). By contrast, our work focuses on settings with unstructured textual input.
>
> We opted not to reimplement their queries for unstructured data, as this would introduce substantial ambiguity due to subjective implementation choices. Instead, we included the PROSE-UnitCost baseline, which aligns with the core assumption of Fish et al. (2024) – namely unit-cost of statements – but uses our own query implementations to ensure comparability.
>
> As a side note, we highlighted that PROSE does not require hyperparameter tuning because it flexibly adapts to varying problem instances without requiring dataset-specific adjustments (such as setting the number of output statements, clustering granularity, or query wording). However, we agree that the term is misleading and we will reword it.
>
> > In other words, I feel like the unit cost version of the model makes more sense, but with the addition that statements should be labeled with the proportion of agents who support them at some meaningful level $\ell$.
>
> We appreciate this suggestion and believe that the answer is dependent on the intended application. Our model is particularly well-suited for settings where the reader’s attention is limited. For instance, in online democratic deliberation platforms (c.f. Bowling Green dataset), larger groups may reasonably expect a more detailed representation of their opinions: That is, a greater share of the limited attention budget of the reader is devoted to articulating their positions in more depth. Moreover, controlling the total slate length while leaving the number of statements flexible is a more general advantage of our approach.
>
> That said, we note that our theoretical framework also applies to the proposed model. Further, PROSE could be adapted to return, along with each statement, a support value, and then use that value to define the cost of the statement.

---

### Official Review · Reviewer_ddgv · 2025-03-10

**Overall Recommendation:** 4

**Summary:**

The paper addresses the task of producing a slat of statements representative of users'opinions. The framework is based on social choice and supported by large language models (LLMs). Theoretical guarantees are provided about the accuracy of the LLM output. The case studies revolve around city improvement measures, drug reviews and showcase the effectiveness of LLMs to generate concise statements representative of users'opinions.

**Update after rebuttal**

I am satisfied with the authors' answer and confirm my positive evaluation of the paper thus recommending its acceptance

**Claims And Evidence:**

Yes. Statements are supported by theoretical proofs showing in support of the accuracy and robustness of their method.

**Essential References Not Discussed:**

not that I am aware of

**Ethical Review Concerns:**

Their application is about social choice but their study can be potentially applied for  unethical reasons. E.g. I would not leave a political decision be taken or summarized by an LLMs. But for what they are showing this could be very much happening. However, I am not saying they are promoting that.

**Ethical Review Flag:**

Flag this paper for an ethics review.

**Ethics Expertise Needed:**

["Responsible Research Practice (e.g., IRB, documentation, research ethics, participant consent)"]

**Experimental Designs Or Analyses:**

Yes. I checked the implementation details in Appendix D

**Methods And Evaluation Criteria:**

yes. The authors integrate LLMs in the pipeline of social choice.

**Other Comments Or Suggestions:**

-In the introduction, "Related Work" paragraph. I invite the author to be more precise in the statement "Another differentiating aspect of our contribution is the focus on deriving mathematical guarantees." Explicitly mention what kind of guarantees you will be deriving.

-In the "General Problem Statement" paragraph I find the notation a bit confusing as you refer to $\mathcal{U}$ as the set of all slates and $u_i$ to the utility of agent $i$—change letter for any of the two.

**Other Strengths And Weaknesses:**

Strengths: providing theoretical guarantees next to languge output.

Weaknesses: not clear to me how to measure the accuracy of the language output.

**Questions For Authors:**

How do you quantify the tolerance and error made by Chat GPT 4o output? How do you measure the accuracy from the language?

**Relation To Broader Scientific Literature:**

I think the impact is very broad and the paper is timely as it related to LLMs and how they can be used in the context of social choice to represent public opinions

**Theoretical Claims:**

No I did not check the proofs in detail.

---

> ### Author Rebuttal · Authors · 2025-03-30
>
> We thank the reviewer for their review.
>
> >  How do you quantify the tolerance and error made by Chat GPT 4o output? How do you measure the accuracy from the language?
>
> The general approach taken in this paper is to design a mechanism (Algorithm 1) that is agnostic to the specific implementation of the underlying queries. Crucially, we show that the mechanism continues to satisfy an approximate notion of proportional representation even when the answers to the queries are subject to error. Importantly, these theoretical guarantees hold without the mechanism needing to know the magnitude of the error in the query answers.
>
> Turning to the empirical evaluation of our implementation using GPT-4o: we assess the quality of the discriminative query implementation in Appendix D.5. Specifically, we use a dataset in which users have written comments and cast upvotes or downvotes on comments written by others. We provide our discriminative query implementation with access to a user’s own comments and task it with predicting how the user rated comments they voted on. This allows us to measure prediction accuracy against known ground truth.
>
> Unfortunately, a direct evaluation of the generative query is more challenging, as it would require knowledge of the optimal statement among a virtually infinite set of possibilities – a task that is infeasible in practice.
>
> Finally, we assess the quality of the resulting slates–which represent the ultimate output of our process–through experimental comparison with baselines (see Table 1). Here, we report (i) the average utility that agents derive from their assigned statement in the slate and (ii) the fraction of BJR violations, which serves as a proxy for the number of underrepresented groups.
>
> > Their application is about social choice but their study can be potentially applied for unethical reasons. E.g. I would not leave a political decision be taken or summarized by an LLMs. But for what they are showing this could be very much happening. However, I am not saying they are promoting that.
>
> Thank you for raising this concern. As we mention in our impact statement, "the use of LLMs to rate and generate statements introduces specific risks that must be carefully addressed before deployment in real-world settings," including "bias," (lack of) "transparency," and "manipulation." This is doubly true in the context of political decision-making, and we will expand the impact statement to make this absolutely clear.

---

### Official Review · Reviewer_MUaS · 2025-03-17

**Overall Recommendation:** 3

**Summary:**

The paper proposes a method for AI assisted democratization, aka using a model to select and aggregate representative candidate statements from social participants.

Main contribution of the work:
- Adding control for summary length instead of number of representative responses, allowing for direct control on the expected cognitive load to read the summary
- Make the system more robust by introducing approximate queries, aka fault tolerant to inaccuracies in utility prediction and popular statement generation
- Experiment demonstrates its effectiveness with GPT-4o as a judge on the user utility, outperforming listed baselines

**Claims And Evidence:**

Claim: The proposed algorithm effectively improves upon previous AI for democracy algorithms.

Evidence:
- Optimal user utility on four datasets (Birth Control-Uniform, Birth Control-Imbalanced, Obesity, Bowling Green), as presented in Table 1.
- The cost-budget analysis framework allows for more direct control on the perceived cognitive workload for consuming the aggregated statement
- Sec 3 proves the near optimality of the algorithm in terms of user utility when the approximated queries are in use

**Essential References Not Discussed:**

n/a

**Experimental Designs Or Analyses:**

See above

**Methods And Evaluation Criteria:**

The paper is well written and easy to follow. Recommending for weak accept for the following reasons:
- The proposed method allows for an extra degree of freedom in controllability (length instead of number of inputs) and shows best result on its benchmark.
- Regarding the evaluation method: Using total user utility as the main metric in evaluation does not immediately come across as the best way to quantify democracy, but neither was I sure how to make it better -- underrepresented groups are still going to be largely neglected in final result generation and their voices deserve to be heard. Also want to learn how the algorithm addresses moral dilemmas such as would it choose to kill one to save a hundred. I understand the design of such metrics would never be perfect but still want to learn the trade offs being considered here and how does the pareto surface look like.
- In AI for governance works we need to think very critically with regard to what kind of value coordinates are we essentially subjecting the AI to because at the end of the day that's the core added value from human if one day say 99% of the work are automated by AIs.

**Other Comments Or Suggestions:**

n/a

**Other Strengths And Weaknesses:**

n/a

**Questions For Authors:**

see above

**Relation To Broader Scientific Literature:**

It's mainly relevant to the following:
- AI for governance
- AI for democracy
- AI for policy making

**Theoretical Claims:**

Been skimming through theorem 3.1-3.4 which ensures near optimality of the proposed algorithm, nothing stands out yet.

---

> ### Author Rebuttal · Authors · 2025-03-30
>
> We thank the reviewer for their review.
>
> > Regarding the evaluation method: Using total user utility as the main metric in evaluation does not immediately come across as the best way to quantify democracy, but neither was I sure how to make it better –underrepresented groups are still going to be largely neglected in final result generation and their voices deserve to be heard.
>
> We fully agree that giving minorities a voice is a fundamental aspect of democratic decision-making. This is precisely why our algorithm is explicitly designed with proportional representation in mind – if a group of agents constitutes x% of the electorate, it should be able to exert control over x% of the slate. For example, if the budget allows for 200 words and we have 100 agents, even a minority group of 5 agents will control 10 words, thereby ensuring that a brief summary of their perspective will be present in the final slate.
>
> To formally capture this ideal, we adopt the axiom of Balanced Justified Representation (BJR). Our theoretical analysis demonstrates that our method satisfies an approximate form of this axiom, even under noisy query implementations (Theorem 3.2). In our experimental evaluation, we measure both total utility but also the frequency of BJR violations (see Table 1, fourth column), which can be interpreted as the fraction of groups whose voices are not adequately reflected in the produced slate, i.e., they are "underrepresented". Across all datasets, PROSE consistently yields fewer BJR violations than the baselines, empirically confirming its ability to give a proportional voice to all groups.
>
> > Also want to learn how the algorithm addresses moral dilemmas such as would it choose to kill one to save a hundred. I understand the design of such metrics would never be perfect but still want to learn the trade offs being considered here and how does the pareto surface look like.
>
> Our algorithm does not impose any specific normative stance on moral dilemmas. Instead, it aims to proportionally summarize the diversity of user opinions, even if these are mutually contradictory. For instance, if the electorate is divided on whether it is ever justifiable to harm one person to save many, the resulting slate might include both: “Harming a person is never ethically justified.” and “Saving many can justify harming an individual.” In this sense, our method captures the full spectrum of views rather than resolving ethical trade-offs by adhering to a particular notion.
>
> > In AI for governance works we need to think very critically with regard to what kind of value coordinates are we essentially subjecting the AI to because at the end of the day that's the core added value from human if one day say 99% of the work are automated by AIs.
>
> We fully share the concern about normative alignment in AI systems. In our setting, however, PROSE is not designed to encode or enforce specific value judgments; rather, it is a tool for faithfully summarizing the input opinions. In particular, the quality and normative content of the output slate largely depend on the user-provided statements. (Broader concerns regarding bias in the underlying LLMs used for query answering are important and discussed in our impact statement.)

---

### Decision · Program_Chairs · 2025-05-01

**Decision:**

Accept (oral)

**Comment:**

The paper deals with "generative social choice" (Fish et al., EC 24), where instead of restricting the set of candidates to the specific statements submitted by participants, we view every possible statement as a potential candidate. This paper extends Fish et al. in a non-trivial manner, namely to include costs and budgets and approximate queries. The authors' rebuttal clarified several issues the reviewers raised, and I hope the authors will address them in their revision.

Overall, the reviewers were positive about this paper, and I share their enthusiasm. I recommend accepting this paper.